# Enhancing Sheep Rumen Function, and Growth Performance Through Yeast Culture and Oxalic Acid Supplementation in a Hemicellulose-Based Diet

**DOI:** 10.3390/microorganisms13122834

**Published:** 2025-12-12

**Authors:** Natnael D. Aschalew, Jialei Liu, Yuetong Liu, Wuwen Sun, Guopei Yin, Long Cheng, He Wang, Wanzhu Zhao, Longyu Zhang, Ziyuan Wang, Huaizhi Jiang, Tao Wang, Guixin Qin, Yuguo Zhen, Zhe Sun

**Affiliations:** 1College of Animal Science and Technology, Jilin Agricultural University, Changchun 130000, China; ednatnael@gmail.com (N.D.A.); sunwuwen@borui.com (W.S.); yinguopei@hotmail.com (G.Y.); 18946506047@163.com (L.C.); wjiajia9993@163.com (H.W.); zuzu13894126994@163.com (W.Z.); 18743070832@163.com (L.Z.); wzy15947868930@163.com (Z.W.); jianghz6806@126.com (H.J.); cagewang@163.com (T.W.); qgx@jlau.edu.cn (G.Q.); 2College of Agriculture and Natural Resources, Dilla University, Dilla P.O. Box 419, Ethiopia; 3College of Life Sciences, Engineering Research Center of Bioreactor and Pharmaceutical Development, Ministry of Education, Jilin Agricultural University, Changchun 130000, China; shicdh@163.com (J.L.); 15946733426@163.com (Y.L.); 4Postdoctoral Scientific Research Workstation, Feed Engineering Technology Research Center of Jilin Province, Changchun Borui Science & Technology Co., Ltd., Changchun 130000, China

**Keywords:** oxalic acid, yeast culture, bacterial composition, rumen fermentation, sheep growth, serum, metabolome

## Abstract

Yeast culture (YC) is a microbial product that enhances ruminal fiber breakdown and improves nutrient digestion and utilization. Our previous research showed that oxalic acid (OA) is a crucial metabolite in YC that enhances rumen function. This study aimed to investigate the effects of YC, OA, and their combination (YO) on rumen function, growth, and fattening in sheep. Twenty lambs were divided into 4 groups (ctrl, YC, OA, and YO; *n* = 5 each) and fed a diet supplemented with 2 levels of YC and 2 doses of OA for 60 days in a 2 × 2 factorial design. Growth and fattening performance, rumen microbiome analysis, serum indices and anti-oxidant levels, and metabolomic profiling were performed. Individual supplementation with YC and OA significantly increased the digestibility of dry matter (DM), organic matter (OM), and crude protein (CP) (*p* < 0.001); neutral detergent fiber (NDF) (*p* < 0.05); and acid detergent fiber (ADF) (*p* < 0.001) and their interaction significantly increased dry matter intake (DMI) (*p* = 0.05). Serum IgA and IgM levels were higher in the supplemented groups (*p* < 0.05). Serum calcium levels were higher in the OA and YO groups (*p* < 0.001). The supplemented groups showed significantly higher growth hormone and superoxide dismutase levels (*p* < 0.05). The longissimus dorsi muscle had higher levels of iron in the OA and YO groups; zinc in the OA, YO, and YC groups (*p* < 0.01); and selenium in the YC group (*p* < 0.05). The OA group had a higher total antioxidant capacity. All supplemented groups showed higher bacterial richness and diversity. *Ruminococcus*, *Succinivibrio*, and *Fibrobacter* were positively correlated with the fermentation and digestibility parameters. The supplementation also altered metabolite levels and types in key physiological pathways. In conclusion, this supplementation improved bacterial composition, nutrient digestibility, weight gain, carcass weight and quality, serum indices, antioxidant levels and metabolomic profiles. This suggests potential for the development of dietary supplements for ruminants.

## 1. Introduction

Sheep and other ruminants rely on microbial fermentation in the rumen to digest plant cell wall components and produce high-quality proteins for human consumption [1]. Cellulose and hemicellulose breakdown is essential for sheep rumen health and productivity [2]. The increasing cost of ruminant feed worldwide highlights the importance of improving the breakdown and fermentation of polysaccharides mainly the aforementioned cell wall components to enhance rumen function and growth in sheep. This can increase productivity and cost efficiency. Hence, improving the breakdown of these cell wall components using eco-friendly additives is crucial.

Yeast culture (YC; *Saccharomyces cerevisiae*), a microbial product, is being studied as a feed supplement for ruminants owing to its potential to improve rumen microbial populations and fermentation [3]. In recent years, metabolomic studies have been increasingly used to explore and deepen insights into research findings. Metabolites are important indicators of an animal’s productivity and overall health [4]. The YC metabolites vary depending on the product and fermentation conditions. Fermentation of molasses produces alcohols, peptides, organic acids, and esters [5]. Additionally, adding organic acids to roughage feed can enhance rumen efficiency by helping maintain a relatively higher pH, balance ammonia nitrogen, increase microbial protein synthesis and VFA production, and decrease methane production [6]. Oxalic acid (OA) is a common organic acid found in the environment. It is produced by a variety of organisms, including fungi, bacteria, plants, and animals [7]. Oxalic acid, when combined with other ingredients, forms oxalate in tropical forages which can negatively impact the health of ruminants by binding to essential minerals such as Ca, Mg, and trace minerals such as Fe, rendering them unavailable to the body [8]. High levels of oxalate in the blood can lead to the formation of crystals that can block urine flow and potentially cause kidney failure [7]. Supplementing with minerals like calcium and magnesium can help reduce the risk of oxalic acid by binding oxalates in the gut, supporting oxalate-degrading microbes in the rumen, preventing mineral deficiencies, and reducing kidney crystal formation. Using this acid as a dietary supplement is crucial for farmers, feed manufacturers, and chemical suppliers [9]. Our previous study has shown that OA is an important metabolite in YC that improves ruminal fermentation efficiency [10]. YC and OA can enhance the digestibility of hemicellulosic diets in the rumen [9,11]. YC promotes fiber-digesting microbes, whereas OA stabilizes the rumen pH, thereby creating ideal conditions for cellulolytic activity and the regulation of microbial populations [9].

Hence, we hypothesized that these supplements might affect rumen function, growth, and fattening performance in sheep. This study aimed to investigate the effects of dietary YC and OA supplementation on rumen function and sheep performance to improve productivity. Furthermore, combining YC and OA in sheep diets may mitigate the risks associated with oxalic acid. Therefore, a thorough evaluation is needed to determine how these supplements influence rumen function, growth, fattening, and the overall health of sheep.

## 2. Materials and Methods

### 2.1. Animal Ethics Statement

All experiments followed the guidelines of the Experimental Animal Ethics Committee of Jilin Agricultural University (JLAU-ACUC2022-003; Changchun, China). The experiment took place in Changchun, Jilin Province, China, from September to November 2023.

### 2.2. Experimental Treatments and Animal Care

Yeast culture was produced in the controlled microenvironment of the JLAU-Borui Dairy Science and Technology R&D Center, whereas the OA dihydrate (C_2_H_2_O_4_•2H_2_O, 99.6% purity) was purchased from RHAWN reagents manufacturing company (Shanghai, China) [9]. The composition and preparation procedures of YC were described in our previous publication [12]. Based on the results of the previous study [9], a diet containing 17% hemicellulose was chosen and supplemented with 2 doses each of YC (0.0 and 0.625 g/kg DM) and OA (0.0 and 0.4 g/kg of feed DM) in a 2 × 2 factorial design, thereby resulting in four groups. The sheep were fed a diet consisting of pelleted feed as the main component and oat hay (*Avena sativa* L.) as a roughage. The ingredients and chemical compositions of the diet [13] are detailed in Table 1.

Twenty male short-tailed Han lambs, aged 3 months and weighing 29.2 ± 1.69 kg, were adapted to a diet for 15 d. They were then divided into four groups (control [ctrl], YC, OA, and combination [YO]; *n* = 5 each) based on the body weight. The ctrl group received the basal diet, the YC group fed a diet supplemented with 0.625 g/kg yeast culture, the OA group received a diet with 0.4 g/kg oxalic acid, and the YO group received a diet with a combination of 0.625 g/kg yeast culture and 0.4 g/kg oxalic acid (DM basis) twice daily for 60 d. They were housed individually in pens equipped with individual feed and water troughs and maintained at constant room temperature. Clean water and mineral licks were provided *ad libitum*.

### 2.3. Nutritional Composition

The feed and fecal samples were ground and passed through a 1 mm screen using a Wiley mill (Thomas Scientific, Swedesboro, NJ, USA). The nutritional levels were analyzed according to the standard methods of Association of Official Analytical Chemicals [14]. The samples were analyzed using the following methods: oven drying at 103 °C (method 934.01) for 3 h for dry matter, ash determination by igniting the sample at 550 °C in a furnace (method 942.05), and organic matter was determined by subtracting the ash content from 100, crude protein content (N × 6.25) was analyzed using a Kjeldahl method (method 984.13), ether extract analysis by Soxhlet extraction (method 920.39), and starch determination by direct acid hydrolysis (method 920.40) [14]. Calcium (Ca) was measured using the disodium ethylenediaminetetraacetate complexometric titration method (method 927.02), while phosphorus (P) was measured using the ammonium vanadate molybdate colorimetric method (method 965.17) [14]. Fiber profiles were determined using an ANKOM^200^ Fiber Analyzer (ANKOM Technology, Macedon, NY, USA). Sodium sulfite and α-amylase were used for NDF analysis, and sulfuric acid was used for ADF. Hemicellulose content was determined by subtracting the ADF level from the NDF level [15]. The digestible energy values were determined based on information from a Feed Database in China [16,17], ensuring consistent and accurate energy calculations. Metabolizable energy (ME) was then calculated using the formula ME = DE × 0.82 accounting for energy losses. This approach provides reliable estimates of available energy in the feeds [13].

### 2.4. Body Weight Gain, Dry Matter Intake, and Digestibility

Sheep body weight was measured initially and every 15 d from 07:30–08:00 a.m. to calculate the average daily and total body weight gain. Daily diet intake and refusal were recorded to determine dry matter intake and feed conversion efficiency. Total feces were collected four times a day for 5 consecutive days using the direct collection method. On days 55–59, total daily feces were collected, weighed, and 10% (*w*/*w*) of it was mixed with 9 mL of diluted H_2_SO_4_ (10% concentration) and stored at −20 °C. A sub-sample of the feces was dried in an air-forced oven at 65 °C for 72 h, ground, and then analyzed for chemical composition. Finally, total dry matter intake (DMI), total fecal output, and fecal dry matter values collected over 5 days were used to calculate nutritional digestibility by comparing the intake diet’s composition with that of the feces using the following general equation.
$$Digestibility\;(\%)=(\frac{\;Intake\;diet\;composition-feces\;composition}{Intake\;diet\;composition})\times 100$$

### 2.5. Serum Biochemical Indices

Blood samples were collected from the jugular veins on d 0, 30, and 60 at 07:30–08:30 a.m., maintained at 23–25 °C for 1 h, and centrifuged at 835× *g* at 4 °C for 15 min to separate the serum. The supernatant serum was stored at −20 °C until serum index analysis. Serum biochemical parameters and antioxidant levels were measured using ELISA kits supplied by Shanghai MLBIO Biotechnology Co., Ltd. (Shanghai, China).

Serum creatinine and calcium levels were analyzed through Qingdao Stantec Standard Testing Co., Ltd, china. using an inductively coupled plasma mass spectrometer (ICPMS) biomedical instrument. To measure Serum creatinine concentration, the process involved diluting the serum with an acidic solution and adding an internal standard. The sample was then analyzed in the ICP-MS, where the plasma components were converted into ions. The instrument detected specific signals derived from creatinine and compared them with calibrated standards. Signal intensities were used to calculate creatinine concentration, and accuracy was verified using quality-control samples. Calcium analysis involved digesting the sample with ultrapure nitric acid, adding an internal standard (aluminum), and diluting the sample for analysis. Collision/reaction cells with gas such as helium were used to minimize interferences for specific calcium isotopes.

### 2.6. Metabolomics Procedures

Metabolic and bacterial composition profiles were analyzed by Nanjing GenePioneer Co., Ltd. (Nanjing, China) using standardized metabolomic data from sheep serum, muscle, and rumen. Data quality control was performed with SIMCA 17 software for PCA [18]. Raw data preprocessing involved filtering unidentified substances and normalizing to an internal standard [19]. Internal standard (IS) solutions were prepared and stored at 4 °C. For internal standard solutions: MSG IS1: 10.0 ± 0.5 mg of malonic acid d2, succinic acid d4, and glycine d5 were dissolved in 10 mL of water in a 15 mL centrifuge tube. CFT IS1: 10.0 ± 0.5 mg of citric acid d4, d-fructose 13C6, and l-tryptophan d5 were dissolved in 10 mL of water in a 15 mL centrifuge tube. LA IS1: 10.0 ± 0.5 mg of l-lysine d4 and l-alanine d7 were dissolved in 10 mL of water in a 15 mL centrifuge tube. SBO IS1: 10.0 ± 0.5 mg of stearic acid d35, benzoic acid d5, and octanoic acid d15 were dissolved in 10 mL of methanol in a 15 mL centrifuge tube. To prepare IS2 Solution: 2 mL of each IS1 stock solution (MSG IS1, CFT IS1, LA IS1, and SBO IS1) were combined, and 4.0 mL of water was added to achieve a final volume of 12.0 mL daily with a nominal concentration of 0.167 mg mL^−1^.

The analyses were performed using a liquid chromatography-mass spectrometry (LC-MS) equipped with a BEH column (Waters ACQUITY UPLC BEH Column 1.7 µm, 2.1 mm × 100 mm) in positive and negative ion modes. Metabolite annotation was performed using Perl in the HMDB and Kyoto Encyclopedia of Genes and Genomes (KEGG) compound databases. Multivariate statistical analyses were conducted in SIMCA using PCA and orthogonal partial least squares discriminant analysis (OPLS-DA). Univariate statistical analyses were carried out using Student’s t-tests and fold change (FC) calculations. Differential metabolite screening and functional analysis were conducted using the Perl programs and the KEGG pathway database [20]. Perl programs were used for differential metabolite screening to detect significant changes between experimental groups and for functional analysis to associate metabolites with metabolic pathways and functions.

### 2.7. Slaughter Samples

#### 2.7.1. Meat Quality Traits

We measured back fat thickness between the twelfth and thirteenth ribs on the left side of the carcass using a vernier caliper and calculated the girth rib (GR) values. The GR measurement is taken by measuring the circumference around the sheep’s chest, just behind the front legs and withers. The longissimus dorsi muscle was used to evaluate meat quality. The longissimus dorsi muscle was removed from the carcass post-slaughter, and its meat color was measured using a fully automatic colorimeter. The colorimeter was placed in close contact with the cross-section of the muscle, and the values (L*, a*, b*) were measured three times for each sheep and averaged. The meat samples were sent to Jiangsu Sambi Biotechnology Co., Ltd. (Nanjing, China) for fatty acid and chemical composition analyses. To determine the fatty acid content, a 0.5 g meat slice was mixed with chloroform and sodium chloride solution and centrifuged, and the lower solution was collected. Dichloromethane was added twice, and the solvent was evaporated. A methanol solution with sulfuric acid was heated, cooled, and n-hexane and aqueous solutions were added. After centrifugation, the supernatant was collected, dried, and an isooctane solution was added for analysis using gas chromatography coupled with mass spectrometry. The chemical composition of the meat was analyzed using the following methods: protein (Kjeldahl nitrogen analyzer, GB5009.5-2016), fat (acid hydrolysis method, GB5009.6-2016), and minerals (optical emission spectrometer—OES) [21].

#### 2.7.2. Fermentation Profiles

On d 61, after a 17 h fasting period, the sheep were weighed and slaughtered. Girth-rib fat thickness (GR) was measured using Vernier calipers. Slices of the longissimus dorsi muscle were frozen in liquid nitrogen and stored at −80 °C until chemical composition and metabolite analyses. Liquids from different parts of the rumen and small intestine were collected and preserved at −20 °C for NH_3_-N and VFA, and at −80 °C for microbial composition and metabolome analysis. Rumen pH was measured in the upper, middle, and inner rumen using a Sanxin MP523-04 digital pH meter (Shanghai Sanxin Instrumentation, Inc., Shanghai, China) according to the manufacturer’s guidelines. Volatile fatty acids were analyzed using an Agilent 7890 A (Agilent Technologies Inc., Santa Clara, CA, USA) gas chromatograph equipped with a 50 m CP-Wax Chrompack silica-fused capillary column. The oven temperatures ranged from 65 to 195 °C, with helium serving as the carrier gas. The detector and injector temperatures were set at 250 °C, with a 1 μL injector volume. After collecting rumen liquid, 0.2 mL of metaphosphoric acid solution (3% *w*/*v*) was added, and the samples were centrifuged at 9391× *g* at 4 °C for 10 min. After refrigeration for 3 h, the sample was centrifuged again to obtain the supernatant. A 1 mL aliquot of the supernatant was then analyzed using gas chromatography. The NH_3_-N concentration was determined using the Chaney and Marbach [22] method, with a Cary Series UV-Vis Spectrophotometer connected to a computer running Cary Series analysis software (v5.0.0.1008, Agilent Technologies, Santa Clara, CA, USA). The detailed procedures are outlined in our previous publication Natnael et al. [9].

### 2.8. Rumen and Intestinal Bacterial Community Composition

#### 2.8.1. 16 S rRNA Gene Sequencing and Taxonomic Composition Classification

The bacterial composition profiles were outsourced to Nanjing GenePioneer Co., Ltd. (Nanjing, China). Rumen and intestinal fluid samples were frozen in liquid nitrogen and stored at −80 °C. Microbial DNA was isolated from the rumen and intestinal fluid samples using a DNA extraction kit (Omega Bio-Tek, Norcross, GA, USA). The DNA content of each sample was quantified using a Nanodrop Spectrophotometer (Thermo NanoDrop 2000). The data were processed using the DADA2 method, which included primer removal, mass filtering, denoising, splicing, and chimera removal. Quality control was conducted by measuring DNA concentration, and the default database used for the 16S rRNA gene was SILVA 138 rRNA database [23]. To determine the taxonomic information of each amplicon sequence variant (ASV), we aligned its sequence with the SILVA Bacterial Database (http://www.arb-silva.de/, accessed on 15 March 2022). The most abundant sequence was selected as the representative sequence for each ASV. We used the QIIME2 software (2022) with the Classify-sklearn algorithm to annotate species for each ASV, using default parameters and a pre-trained Naive Bayes classifier. Customized primers targeting the 16S V4-V5 region were used to amplify a 420 bp fragment, and sequencing was performed on the Illumina Novaseq6000 platform, generating 2 × 250 bp paired-end data for 16S analysis. The V4-V5 region of the 16S rRNA gene was amplified using primers 515F (5′-GTGCCAGCMGCCGCGG-3′) and 907R (5′-CCGTCAATTCMTTTRAGTTT-3′) by PCR (95 °C for 5 min, followed by 27–30 cycles at 95 °C for 30 s, 55 °C for 30 s, and 72 °C for 45 s and a final extension 72 °C for 15 min). PCR reactions were performed in a 30 µL mixture containing 15 µL of 2 × Phanta Master Mix, 1 µL of each primer (10 µM), and 20 ng of template DNA Amplicons were extracted from 2% agarose gels and purified using the AxyPrep DNA Gel Extraction kit (Axygen Biosciences, Union City, CA, USA). The PCR products were measured using a Qubit^®^3.0 (Life Invitrogen) and subsequently combined in equal proportions. The pooled DNA was then used to create an Illumina PairEnd library following Illumina’s genomic DNA library preparation protocol.

#### 2.8.2. Bacterial Diversity Indices

Alpha diversity analysis was performed using QIIME2, including dilution curve analysis in R (version 3.6.2) [24]. Species richness and diversity were evaluated using the Chao1, Shannon, and Simpson indices. Beta diversity analysis was performed using QIIME2 to compare the species diversity among the samples. PLS-DA was used to analyse variations in the microbial community composition and evaluate how different taxa affected the samples.

### 2.9. Statistical Analysis

The MIXED procedure in SAS (version 9.4; SAS Institute Inc., Cary, NC, USA) [25] was used to analyze variables. A two-way analysis of variance (ANOVA) was performed with treatments as fixed effects and animals as random effects. The significance level for Duncan’s New Multiple Range Test was set at *p* ≤ 0.05 to analyze significant variables in mean comparisons. We used SPSS version 28 (Armonk, NY, USA IBM Corp) to conduct canonical correlation analysis to assess the relationship between bacterial genera and rumen fermentation and growth.

The experimental model was as follows:Y_ijkl_= µ + Y_i_ + O_j_ + A_k_ + (YO)_ij_ + e_ijkl_ where Y_ijkl_, the dependent variables; µ, overall mean; Y_i_, yeast culture effect (_i_ = 0 and 0.625); O_j_, oxalic acid effect (_j_ = 0 and 0.4); A_k_, random effect of animals (_k_ = 1–5); (YO)_ij_, effect due to YC and OA interaction; and e_ijkl_, random residual error.

The sequencing data was processed through various steps, such as sorting samples by barcodes, merging paired reads, quality control, and filtering to ensure high-quality sequences. Data demultiplexing was done using specific mismatch criteria for barcodes and primers. Pandaseq software was used to merge paired reads with defined overlap and mismatch parameters [26]. PRINSEQ software was employed to filter out low-quality bases and sequences with excessive N bases [27].

Metabolomics data analysis was conducted using LC-MS/MS (Orbitrap Exploris 120, Thermo Fisher Scientific, Waltham, MA, USA) [28] with three main components: basic analysis, advanced analysis, and personalized analysis. Basic analysis includes univariate and multivariate statistical analysis to identify significant metabolite differences. Personalized analysis involves bioinformatics analysis of these metabolites. Advanced analysis was customized to meet specific requirements, including multi-omics association analysis and the creation of custom charts for publications [29].

## 3. Results

### 3.1. Nutrient Digestibility and Growth Performance in Sheep

The interaction of YC and OA (YC × OA) significantly increased dry matter, organic matter, crude protein, hemicellulose, starch, ash, and ether extract (*p* < 0.001) digestibility. Individual supplementation of YC and OA also affected NDF and ADF digestibility. The ctrl group had higher fecal Ca levels than the supplemented groups (*p* < 0.001, Table 2).

The YC and OA interaction resulted in significantly higher DMI (*p* < 0.05). The average daily weight gain (ADG) and overall body weight gain (BWG) increased in the YC and OA supplemented groups compared to those in the ctrl group. Individual supplementation with OA increased the feed conversion ratio compared to the ctrl and other supplementation groups (*p* > 0.05). Carcass weight increased in all supplemented groups (*p* > 0.05) (Table 3). The supplemented groups had significantly higher dressing percentages than the ctrl group (*p* < 0.05). Additionally, the YO group had significantly higher girth-rib fat thickness than the ctrl group (*p* < 0.05). The supplemented groups showed a higher abdominal fat content, but there was no significant difference between the groups (Table 4).

### 3.2. Sheep Serum Indices

On d 60, OA supplementation significantly increased total cholesterol (TC) levels (*p* < 0.01), whereas YC and their combination (YO) supplementation decreased TC levels (*p* < 0.01). YC significantly increased glucose (GLU) levels (*p* < 0.05). On d 30, OA group had significantly higher IgA levels than the ctrl group (*p* < 0.05). On d 60, the YO, YC, and OA groups showed higher IgM levels than the ctrl group. The YC and OA groups had significantly higher blood urea nitrogen (BUN) levels than the ctrl and YO groups (*p* < 0.001), whereas the YO group had significantly lower BUN levels than the ctrl group (*p* < 0.05), suggesting that combining YC and OA supplementation may help regulate BUN levels. The creatinine concentration was significantly lower in the YO group. No renal malfunction was detected, as creatinine levels did not differ between the ctrl and OA groups. The OA group showed significantly higher serum Ca levels compared with the other groups (*p* < 0.001). Supplementation with YC, OA, and YO significantly increased superoxide dismutase (SOD) levels on day 60 (*p* < 0.05). The OA group exhibited a significantly higher concentration of total antioxidant capacity (T-AOC) compared to the other groups on day 60 (*p* < 0.05). The YO group significantly increased growth hormone (GH) levels at both 30 and 60 days (*p* < 0.05, Table 5).

### 3.3. Sheep Serum Metabolomics

#### 3.3.1. Metabolite Differences Between Groups

The OPLS-DA model effectively classified and distinguished the two sample groups. The scatter plot of the OPLS-DA scores demonstrates a clear separation between the groups, with all samples falling within the 95% confidence interval. Variations among the six comparisons (ctrl vs. YC, ctrl vs. OA, ctrl vs. YO, YC vs. OA, YC vs. YO, and OA vs. YO) were evident based on the main (t[1]p) and orthogonal (t[1]o) component scores (Appendix A). The permutation test analysis confirmed the absence of overfitting in the model, ensuring the reliability of the results (Appendix A). The distinct effects of YC, OA, and YO supplementation on the metabolites were highlighted by the significant differences observed between each pair of experimental groups.

#### 3.3.2. Identification and Analysis of Key Metabolites in Sheep Serum

The OPLS-DA results were used to identify differential metabolites by combining VIP values with a *p*-value. A metabolite was considered differential if *p*-value was <0.05, VIP was >1, and Fold Change was >1. We found 48, 89, 161, 34, 93, and 98 differential metabolites in the ctrl vs. YC, ctrl vs. OA, ctrl vs. YO, YC vs. OA, YC vs. YO, and OA vs. YO groups, respectively (Figure 1). A total of 98 metabolites were identified in the OA and YO groups, of which 90 were upregulated. Additionally, the top 20 metabolites in sheep serum, along with their VIP scores and *p*-values for different comparison groups, are shown in Appendix A, indicating significant differences and biological relevance.

#### 3.3.3. Analyzing Metabolite Pathway Enrichment in Sheep Serum

KEGG metabolic pathway analysis was used to identify the impact of the supplements on the metabolic pathways. Figure 2A shows the 10 KEGG pathways between the ctrl and YC groups, predominantly lipid and carbohydrate metabolism pathways. Thirteen pathways were identified between the ctrl and OA, including the amino acid, cofactor and vitamin, and lipid metabolism pathways (Figure 2B). Figure 2C shows 27 pathways in the ctrl and YO groups, including the amino acid, carbohydrate, and energy metabolism pathways. Figure 2D shows 28 pathways in the YC and OA comparison groups, including amino acid, nucleotide, and lipid metabolism pathways. Figure 2E shows 19 pathways in the YC and YO groups, including amino acid, carbohydrate, and nucleotide metabolism pathways. Figure 2F shows the predominant pathways in the OA and YO groups, including amino acid, carbohydrate, lipid, and vitamin metabolism pathways.

Additionally, Figure 3A shows the third-level KEGG pathways between the ctrl and YC groups, including C5-branched dibasic acid and glyoxylate and dicarboxylate metabolism, with higher numbers of annotated metabolites and enrichment factors. Figure 3B illustrates the pathways in the ctrl and OA groups, including vitamin digestion, biosynthesis of amino acids and secondary metabolites, and biosynthesis of cofactors with high metabolite numbers and enrichment factors. Figure 3C shows the pathways of the ctrl and YO groups. Metabolic pathways and microbial metabolism are the dominant pathways. Figure 3D shows the pathways in the YC and OA groups, mainly bile secretion and arginine, proline, purine, tropane, piperidine, and pyridine alkaloid biosynthesis. Figure 3E shows the pathways between the YC and YO groups, including C-5 branched dibasic acid, arginine, proline, pyrimidine, and purine metabolism. Figure 3F shows the pathways between the OA and YO comparison groups, including tyrosine and phenylalanine metabolism, and the biosynthesis of phenylalanine, tyrosine, and tryptophan.

### 3.4. Meat Quality Traits

#### 3.4.1. Physicochemical Properties

OA and YO supplementation significantly increased the meat brightness (*p* < 0.001) and redness (*p* < 0.05). YC supplementation significantly increased redness compared to the ctrl group (*p* < 0.05). The fat content of the longissimus dorsi muscle was significantly higher in the supplemented groups than that in the ctrl group (*p* < 0.001). Iron content was significantly higher in the OA and YO groups than in the YC and ctrl groups (*p* < 0.001). The zinc content was significantly higher in the OA group than in the others (*p* < 0.01), and the YC and YO groups also showed significantly higher zinc content than the ctrl group (*p* < 0.01). Selenium was significantly higher in the YC group. However, the ctrl group showed significantly higher P and Ca contents compared to the supplementation groups (*p* < 0.001, Table 6).

#### 3.4.2. Fatty Acid Composition of Longissimus Dorsi Muscle

Twenty-one fatty acids, including saturated fatty acids (SFA), monounsaturated fatty acids (MUFA), and polyunsaturated fatty acids (PUFA) were identified in the muscle tissue. The YC group had higher levels of SFA, MUFA, and PUFA compared to the other groups, with significantly high levels of lauric, oleic, linoleic, and α-linoleic acids (*p* < 0.05). The ctrl group contained higher levels of undecanoic acid and heptadecenoic acid. Muscle fatty acid concentrations decreased in the OA and YO groups (*p* < 0.05), whereas the PUFA to SFA ratio was higher in these groups (Table 7).

### 3.5. Meat Metabolomics

#### 3.5.1. Metabolite Differences in the Longissimus Dorsi Muscle

In Appendix A, A and A’ present the OPLS-DA dispersion and permutation test results for comparison between the ctrl and YC groups, respectively. B and B’ show the OPLS-DA dispersion and permutation test results for comparison between ctrl and OA groups, respectively. C and C’ indicate the OPLS-DA dispersion and permutation test results for the comparison between the ctrl and YO groups, respectively. D and D’ indicate the OPLS-DA dispersion and permutation test results for comparison between the YC and OA groups, respectively. E and E’ show the OPLS-DA dispersion and permutation test results for comparison between the YC and YO groups, respectively. F and F’ indicate the OPLS-DA dispersion plot and permutation test results for comparison between the OA and YO groups, respectively.

#### 3.5.2. Screening and Analysis of Key Metabolites in Longissimus Dorsi Muscle

Figure 4 present the differential metabolites in sheep longissimus dorsi muscle across different groups. Between the ctrl and YC groups, 159 differential metabolites were identified, 40 of which were upregulated. Between ctrl and OA groups, 96 differential metabolites were identified, of which 30 were upregulated. For ctrl and YO groups, 271 differential metabolites were found, of which 49 were upregulated. In the YC and OA comparison, 54 differential metabolites were identified, of which 27 were upregulated. Between YC and OA groups, 214 differential metabolites were found, of which 44 were upregulated. Lastly, 51 differential metabolites were identified in the OA and YO comparison, of which 39 were upregulated. Additionally, Appendix A shows the top 20 metabolites with higher VIP scores in different comparison groups, indicating significant differences and their relevance to biological functions.

#### 3.5.3. Analyzing Metabolite Pathway Enrichment in Longissimus Dorsi Muscle

A total of 22 s-level KEGG pathways were identified in the ctrl and YC groups, with the key pathways including amino acid metabolism, digestive system, and biosynthesis of secondary metabolites (Figure 5A). A total of 24 pathways were identified between the ctrl and OA groups, predominantly related to amino acid, digestive system, and cofactors and vitamins metabolism (Figure 5B). Figure 5C shows the 29 pathways identified in the ctrl and YO groups, with key pathways including carbohydrate, lipid metabolism, and the digestive system. Figure 5D shows the 21 pathways between the YC and OA groups, with lipid metabolism, digestive system, and amino acid metabolism being the dominant pathways. As shown in Figure 5E, 26 pathways were identified in the YC and YO groups, with amino acid, carbohydrate, and lipid metabolisms being the dominant pathways. Figure 5F illustrates the 15 pathways between the OA and YO groups, with carbohydrate, lipid, and nucleotide metabolism being dominant.

Additionally, protein digestion and absorption, amino acid biosynthesis, ABC transporters, 2-oxocarboxylic acid metabolism, and aminoacyl-tRNA biosynthesis pathways showed higher levels of metabolites and enrichment factors in the longissimus dorsi muscles of the ctrl and YC groups (*p* < 0.05, Figure 6A). Vitamin digestion and absorption and valine, leucine, and isoleucine degradation pathways were significantly higher in the ctrl and OA groups (*p* < 0.05, Figure 6B). The pentose phosphate, nucleotide metabolism, glucosinolate biosynthesis, and phenylpropanoids biosynthesis pathways had significantly higher numbers of metabolites and enrichment factors in the ctrl and YO groups (Figure 6C). Glycerolipid metabolism and lysine degradation pathways were annotated by higher numbers of metabolites and enrichment factors in the YC and OA groups (Figure 6D). Glycerophospholipid and nucleotide metabolism were the main pathways with significantly annotated differential metabolites and enrichment factors between the YC and YO groups (Figure 6E). Figure 6F shows the enriched pathways in the OA and YO groups, including glyoxylate and dicarboxylate metabolism, glycerophospholipid metabolism, pyrimidine metabolism, terpenoids and steroids biosynthesis, and sulfur and lipid metabolism.

### 3.6. Rumen Fermentation and Bacterial Composition in Sheep Rumen and Small Intestine

#### 3.6.1. Sheep Rumen Fermentation Parameters

Supplementation with YC, OA, and YO significantly increased rumen pH (*p* < 0.05). YO supplementation also significantly increased the ammonia nitrogen concentration (*p* < 0.05). All the supplemented groups showed significantly higher acetic acid production than the ctrl group (*p* < 0.05). The YC and YO groups had significantly higher concentrations of propionic acid, TVA^3^, and TVFA^6^ than the ctrl and OA groups (*p* < 0.05). Combined supplementation with YC and OA significantly increased the acetic to propionic acid ratio (A/P) compared to that in the ctrl group (*p* < 0.05), whereas individual supplementation with YC decreased the A/P ratio (*p* < 0.05, Table 8).

#### 3.6.2. Bacterial Composition in the Rumen and Small Intestine

•Alpha diversity indices

Chao1 bacterial richness was significantly higher in the supplementation groups (*p* < 0.05), with the YC and YO groups showing a higher richness than the OA group. Good’s coverage for all groups exceeded 99.9%, indicating adequate sequencing depth. Shannon and Simpson diversity indices were significantly higher in the YO group (*p* < 0.05), and the Shannon diversity indices of the YC and OA groups were significantly higher than those of the ctrl group (*p* < 0.05).

This study also found a diverse bacterial community in the small intestine, with a significantly higher richness in the YC group (*p* < 0.05). However, the OA and YO groups showed significantly lower richness than the ctrl group (*p* < 0.05). The Shannon and Simpson diversity indices increased in the supplemented groups, but the differences were not statistically significant (Table 9).

•Beta diversity

The results of PLS-DA for rumen and intestinal fluids are shown in Figure 7A,C, respectively. The supplementation and ctrl groups were clearly different from each other.

The Bray–Curtis beta diversity distance index showed significant differences between the ctrl and YC (*p* < 0.001), ctrl and OA (*p* < 0.01), and ctrl and YO (*p* < 0.05) groups in the rumen (Figure 7B). In the small intestine, significant differences were observed between the ctrl and YC (*p* < 0.01), ctrl and OA (*p* < 0.05), and ctrl and YO (*p* < 0.01) groups (Figure 7D).

•Bacterial phylum composition in the rumen and small intestine

YC and OA supplementation altered the composition of bacterial phyla in the rumen and small intestine of the sheep (Figure 8A,B). A total of 24 phyla were identified in the rumen fluid. Fourteen of these were common to all groups; one was shared with the ctrl, YC, and YO groups; one was found in the YC, OA, and YO groups; one was shared by the YC and OA groups, and the remaining six were unique to the OA group (Figure 8E). The dominant bacterial phyla in the rumen fluid were Bacteroidetes, Firmicutes, Proteobacteria, Spirochetes, and Fibrobacterota. The YC, OA, and YO groups increased the phyla Bacteroidetes, Spirochetes, Fibrobacterota, and Actinobacteria compared to the ctrl group, although the differences were not statistically significant (*p* > 0.05). Forty bacterial phyla were found in the small intestine, 22 of which were common to all groups. The ctrl group had three unique phyla, and the YC group had two unique phyla (Figure 8F). The dominant phyla in the small intestine are Firmicutes, Actinobacteria, Proteobacteria, Bacteroidetes, and Cyanobacteria. Firmicutes showed a higher relative abundance in the supplemented groups (*p* < 0.05). The relative abundance of Proteobacteria, Bacteroidetes, and Cyanobacteria was higher in the YC group, whereas that of the phylum Elusimicrobiota was higher in the OA group (*p* > 0.05).

•Bacterial genus composition in the rumen and small intestine

Supplementation with YC, OA, or YO altered the composition of bacterial genera in the rumen and small intestine of sheep (Figure 8C,D). A total of 232 genera were identified in the rumen, with 100 genera common to all groups. The OA, YC, YO, and ctrl groups had 44, 23, 13, and 2 unique genera, respectively. The remaining genera were shared among different groups (Figure 8G).

*Prevotella*, *Rikenellaceae RC9 gut group*, *Succinivibrio*, and *Ruminococcus* were among the dominant genera identified in the rumen (Figure 8C). The relative abundances of *Prevotella*, *Ruminococcus*, and *Fibrobacter* increased in the YC-, OA-, and YO-supplementation groups compared to the ctrl group (*p* > 0.05). In total 514 genera were identified in the small intestine, 118 genera were common to all groups; 91 unique to the YC group, 62 unique to the ctrl group, 29 unique to the OA group, and 29 unique to the YO group. The remaining genera were shared among different groups (Figure 8H).

The dominant genera in the small intestine were *Olsenella*, *[Eubacterium] coprostanoligenes group*, *Mycoplasma*, *RF39*, and *[Ruminococcus] gauvreauii group*. The abundance of the *Olsenella* genus was significantly higher in the ctrl group (*p* < 0.05), whereas *Mycoplasma*, *RF39*, and *[Ruminococcus] gauvreauii group* were higher in the YC, OA, and YO groups. The *[Eubacterium] coprostanoligenes group* genus was significantly higher in the OA and ctrl groups compared to that in the other groups. The abundance of the *Syntrophococcus* genus was higher in the OA group, although this difference was not statistically significant. *Acetitomaculum* and *Lachnospiraceae NK3A20 group* genera were more abundant in the OA group. The relative abundance of the *Lachnospiraceae NK3A20 group* genus was significantly higher in the YC group (Figure 8D).

•Analysis of taxon differences between groups

The linear discriminant analysis (LDA) effect size (LEfSe) method (*p* < 0.05, LDA score > 3) was used. In the rumen, the YO group showed significantly higher relative abundances of *Prevotella ruminicola* and *Treponema ruminis*. The YC group showed significantly higher levels of *Paraprevotella*, and *Izemoplasmatales*. The OA group exhibited significantly higher levels of *Prevotellaceae-Ga6A1-group* and *Suttonella*. In the ctrl group, only *Shuttleworthia* exhibited significantly higher levels (Figure 9A). In the intestinal fluid, the Actinobacteriota phylum, and *Olsenella*, *Propionibacterium*, *Ruminobacter*, and *Actinomyces* genera were significantly higher in the ctrl group. The abundance of Romboutsia and Ureaplasma phyla was significantly higher in the YO group. Additionally, *Mycoplasma*, *Pseudoxanthomonas*, *Ramlibacter*, and *Nesterenkonia* were significantly more abundant in the YC group (Figure 9B).

### 3.7. Correlations Between Bacterial Genera and Rumen Fermentation and Growth in Sheep

Rumen pH was positively correlated with the abundance of *Prevotella* and *Succinivibrio*. Acetic acid was strongly correlated with *Ruminococcus* (*p* < 0.05), *p2534*, and *F082*. Butyric acid showed a strong positive correlation with *F082* (*p* < 0.05) and *Fibrobacter* (*p* < 0.01) abundance. The acetic to propionic acid ratio was positively correlated with *Ruminococcus*. The digestibility of DM, organic matter, crude protein, neutral detergent fiber, acid detergent fiber, and starch was positively correlated with *Rikenellaceae RC9 gut group*, *Rikenellaceae*, *Succinivibrio*, and *Succinivibrionacea* (*p* > 0.05, Figure 10).

### 3.8. Effects of YC and OA Supplementation on Sheep Rumen Metabolome

#### 3.8.1. Rumen Metabolite Differences Between Groups

The OPLS-DA model effectively classified and distinguished the two sample groups. The scatter plot of the OPLS-DA scores demonstrates a clear separation between the groups, with all samples falling within the 95% confidence interval. Variations between the six groups (ctrl vs. YC, ctrl vs. OA, ctrl vs. YO, YC vs. OA, YC vs. YO, and OA vs. YO) were evident based on the main (t[1]p) and the orthogonal component (t[1]o) scores (Appendix A). The permutation test analysis confirmed the absence of overfitting in the model, ensuring the reliability of the results (Appendix A). The distinct effects of YC, OA, and YO supplementation on the metabolites were highlighted by the significant differences observed between each pair of experimental groups.

#### 3.8.2. Screening and Analysis of Key Metabolites in Sheep Rumen

In total, 137, 449, 273, 68, 68, and 14 differential metabolites were identified in the ctrl vs. YC, ctrl vs. OA, ctrl vs. YO, YC vs. OA, YC vs. YO, and OA vs. YO groups, respectively. Figure 11 shows 137 differential metabolites in the ctrl and YC-supplemented groups, with 10 being upregulated. Between the ctrl and OA groups, 449 differential metabolites were identified, 14 of which were upregulated. A total of 273 differentially expressed metabolites were identified in the ctrl and YO groups, with 21 upregulated and the remaining downregulated. In the YC vs. OA comparison, 68 differential metabolites were identified, of which 2 were upregulated. Among the YC- and YO- supplementation groups, 68 differential metabolites were identified, of which only 2 were upregulated. In the OA vs. YO comparison, 14 differentially expressed metabolites were identified, of which 12 were upregulated and 2 were downregulated.

Additionally, Appendix A shows the top 20 metabolites in the rumen fluid, their VIP scores, and *p*-values for different comparison groups. Higher VIP scores suggested that the corresponding metabolites played a more critical role in distinguishing between groups in the biological functions of rumen bacteria.

#### 3.8.3. Analyzing Metabolite Pathway Enrichment in Sheep Rumen

Figure 12A shows 13 s-level KEGG pathways enriched with metabolites in the ctrl and YC groups, with amino acid metabolism and xenobiotics biodegradation metabolism being dominant. Thirty-one KEGG pathways were identified in the ctrl and OA groups. These included amino acid metabolism, carbohydrate metabolism, biosynthesis of secondary metabolites, nucleotide metabolism, and the digestive system (Figure 12B). Nineteen KEGG pathways were identified in the ctrl and YO groups, with nucleotide, amino acid, lipid, and carbohydrate metabolism being the dominant pathways (Figure 12C). Figure 12D shows 13 KEGG functional pathways between the YC and OA groups, including carbohydrate, amino acid, lipid, and nucleotide metabolism pathways. Twelve KEGG pathways were identified between YC and YO, with nucleotide metabolism and the biosynthesis of secondary metabolites being the most prominently annotated with differential metabolites (Figure 12E). Figure 12F shows the seven KEGG pathways that were differentiated between the OA and YO groups. These pathways included carbohydrate, nucleotide, and energy metabolism pathways.

Figure 13 shows the top 20 third-level pathways enriched with differential metabolites between different groups. The biosynthesis of various secondary metabolites, antibiotics, glycine, serine, and threonine metabolism, nucleotide metabolism, and microbial metabolism pathways were enriched with more metabolites in the ctrl and YC groups (Figure 13A). Nucleotide metabolism, amino acid biosynthesis, aminoacyl-tRNA biosynthesis, and 2-oxocarboxylic acid metabolism had higher numbers of metabolites and enrichment factors in the ctrl and OA groups (*p* < 0.05; Figure 13B). The major pathways in the ctrl and YO groups were pyrimidine, nucleotide, and purine metabolism (*p* < 0.05, Figure 13C). The key pathways in the YC and OA groups were microbial metabolism in diverse environments, glyoxylate and dicarboxylate metabolism, cysteine and methionine metabolism, purine metabolism, the TCA cycle, and C5-branched dibasic acid metabolism (Figure 13D). Caffeine, purine, and nucleotide metabolism showed significantly higher levels of metabolites and enrichment factors in the YC and YO groups (*p* < 0.05; Figure 13E). Figure 13F illustrates the pathways in the OA and YO groups, including metabolic pathways, microbial metabolism in diverse environments, and C5-branched dibasic acid metabolism.

### 3.9. Functional Prediction Analysis

YC supplementation improved amino acid and carbohydrate metabolism, as well as secondary metabolite biosynthesis. OA supplementation enhanced transcription and carbohydrate metabolism. YO supplementation boosts lipid, carbohydrate, and xenobiotic biodegradation. The control group exhibited improvements in glycan metabolism and immune system pathways in the rumen (Figure 14A). Figure 14B shows the predicted functional pathways of rumen fluid in each sample.

## 4. Discussion

### 4.1. Nutrient Digestibility and Growth Performance in Sheep

Improving feed intake and nutrient digestibility in ruminants is essential for enhancing growth and productivity and reducing production costs [30]. Similar to our study, Dai et al. [31] reported that YC supplementation in dairy cattle increased nutrient digestibility, suggesting that YC and OA supplementation improved nutrient digestibility and mineral absorption, leading to improved growth performance through enhanced nutrient utilization. The improved nutrient digestibility is likely due to alterations in the rumen microbiome composition. Increasing the DMI in sheep can enhance nutrient digestibility, feed efficiency, weight gain, carcass quality, and rumen fermentation efficiency, ultimately improving growth performance [32]. In this study, YC and OA supplementation increased ADG and BWG. This could be attributed to better nutrient utilization, enhanced gut function, or improved microbial balance, resulting in more efficient conversion of feed. Wang et al. [33] found that the addition of YC to sheep diets increased weight gain and feed efficiency. Furthermore, the higher carcass weight suggests that YC and OA can increase the overall meat production. Abu El-Kassim et al. [34] also reported similar findings.

### 4.2. Serum Biochemical Indices and Metabolome

The increased TC levels in OA group, may be due to the increased feed intake and digestibility, leading to higher fat deposition in the body and bloodstream. TC and triglycerides (TG) are fat molecules present in the bloodstream that provide energy. Volatile fatty acids are absorbed in the rumen through passive diffusion across the ruminal epithelium into the bloodstream. The remaining passage through the small intestine with higher absorption may increase TC and TG concentrations in the bloodstream [35]. Increased TC and TG levels can affect plasma lipid concentrations, potentially influencing lipid metabolism and distribution, particularly in the lipoprotein fractions [36], and promoting reproductive performance in sheep [35]. The YC group showed higher serum glucose levels, which is in line with the findings of Tavares et al. [37], who reported a similar increase in sheep. Higher levels of IgA, IgG, and IgM in the OA and YO groups indicating potential immune system enhancement in sheep. Additionally, lower BUN levels in the YO group suggesting that combining YC and OA supplements may help regulate BUN levels. No renal malfunction was detected, as creatinine levels did not differ between the ctrl and OA groups. Contrarily, Su et al. [38] reported that supplementing a diet with YC reduced the plasma urea nitrogen level in sheep. We found that OA supplementation significantly increased serum calcium levels, in contrast to previous studies by Duncan et al. [7] and Rahman et al. [39] who reported that OA decreased serum Ca levels.

In the industrial era, the release of free radicals into the environment has become a significant threat. Animals are constantly exposed to the reactive oxygen species (ROS) generated by environmental chemicals and metabolic processes, necessitating measures to mitigate their harmful effects. Antioxidants help detoxify these free radicals and maintain animal health. An imbalance between antioxidants and ROS leads to oxidative stress [40]. Antioxidants can reduce DNA damage, increase enzymatic activity, and reduce stress in sheep [41]. The increase in superoxide dismutase (SOD) in the supplemented groups may indicate a potential defense against the harmful effects of free radicals in the body and response to oxidative stress or inflammation [42]. The OA group showed an increased total antioxidant capacity, suggesting a stronger defense against free radicals and oxidative stress [43]. Supplemented groups showed higher growth hormone levels, potentially enhancing the growth performance of the sheep.

Glyoxylate and dicarboxylate metabolism pathway plays a role in energy production during nutrient intake when the oxygen supply is limited [44], and the C5-branched dibasic acid metabolism pathway is associated with amino acid metabolism, converting acetyl-CoA and pyruvate to 2-methyl succinic acid [45]. It enhances the absorption of vitamins that are essential for nutrient intake and microbial vitamin synthesis [46], whereas purine metabolism is involved in nucleotide synthesis, microbial protein synthesis, and nitrogen utilization efficiency [47]. Pentose and glucuronate interconversions are important for producing various biomolecules and cellular energy by generating a high NADPH redox potential and enhancing nucleotide and nucleic acid biosynthesis [48]. Kahraman et al. [49] found that tryptophan metabolism; phenylalanine, tyrosine, and tryptophan biosynthesis; and phenylalanine, cysteine, and methionine metabolism were the most dominant metabolic pathways in fattening sheep.

### 4.3. Effects on Meat Quality

Sheep meat production involves both traditional and modern systems, with various processes affecting the meat quality [50]. Feed intake and digestibility play crucial roles in determining the quality of meat. YC and OA supplementation improved meat brightness and redness. Similarly, YC increased meat redness in lambs [51] and in pigs [52]. However, Suryaningsih et al. [53] reported that YC had no effect on meat color in sheep. Supplementation with YC, OA, and YO improved meat quality by increasing fat content and essential mineral availability (Fe, Zn, Se) in the longissimus dorsi muscle. These minerals are important for human health [54]. Similarly, Milewski and Zaleska [51] reported that adding YC to the diet of lambs increased fat content in the longissimus lumborum muscle. A high selenium content in meat may also indicate a strong immune status in animals [55]. Red meat is a rich source of protein, vitamins, and minerals, but also contains saturated fatty acids that can be harmful to health, leading to cardiovascular disease and other health problems [56]. The OA and YO groups reduced the saturated fatty acids (SFA) content in the longissimus dorsi muscle, potentially improving the healthiness of the meat. Reducing SFA in meat can improve its tenderness, flavor, and overall quality of the meat [57].

Molecular meat quality measurements, particularly metabolomics, are increasingly used to better understand biochemical changes in meat [4]. Degradation of the valine, leucine, and isoleucine pathways is significantly influenced by microbial composition in the rumen. This has a direct effect on the availability of these amino acids in meat, which, in turn, affects its nutritional quality [58]. Glascock et al. [59] also found that the glycerophospholipid pathway significantly influenced the lipid composition of camel and beef. This pathway is involved in the production and breakdown of phospholipids in muscle tissue. Phospholipids are crucial for the cell membrane structure and have various functions in the body [60]. The addition of YC and OA to sheep’s diet altered meat metabolites and metabolic pathways, increasing arachidonic acid, an essential omega-6 fatty acid precursor. This acid regulates inflammation, immune response, and physiological functions in sheep meat, impacting its quality, tenderness, and flavor [61]. The glycerophospholipid pathway showed significant enrichment in supplemented groups, indicating its importance in lipid composition and metabolism. The YC and OA supplementation increased fat content in the longissimus dorsi muscle, suggesting a potential contribution to the increased enrichment of this pathway. The longissimus dorsi muscle in the supplemented groups primarily contains protein digestion and absorption, biosynthesis and degradation of amino acids, vitamin digestion and absorption pathways. These pathways are crucial for the availability of amino acids in the meat, which in turn determines the protein content [62].

### 4.4. Effects on Rumen Fermentation

Maintaining a normal pH can enhance sheep growth performance in intensive production systems, optimize microbial composition, and protect the host from subacute ruminal acidosis. This supplementation increased the ruminal pH, which can have positive implications for rumen health and fermentation efficiency. Fabino et al. [63] reported that sheep with a higher ruminal pH showed higher nutrient degradation owing to the increased colonization of microbes in the rumen. A higher concentration of ammonia nitrogen in the rumen may indicate an excess amount produced through microbial synthesis and nutrient utilization. This study also showed an increased concentration of acetic, propionic, and TVFA in the supplemented groups. This supplementation improved nutrient digestion, absorption, and growth performance [64]. Increasing the ratio of acetic to propionic acids can improve dietary nitrogen utilization and growth performance.

### 4.5. Bacterial Phylum and Genus Composition

Analysis of bacterial composition helps assess beneficial and pathogenic bacteria levels and identify bacterial taxa associated with efficient feed conversion and methane emission, which are key factors in ruminant production [65]. The rumen microbiome significantly affects the nutritional, physiological, and immunological states of the host, directly affecting overall health and production performance [66]. The Bacteroidetes phylum includes bacterial species that ferment complex carbohydrates, including hemicellulose and cellulose [67]. Spirochetes and Fibrobacterota are phyla that include species of complex carbohydrate-degrading bacteria [66]. In the small intestine, the most dominant phyla were Firmicutes, Actinobacteria, and Proteobacteria. Similarly, Wang et al. [68] found that Firmicutes and Proteobacteria are the dominant phyla in the small intestine of sheep. The intestinal microbiota is essential for breaking down slowly digestible plant fibers, non-fiber carbohydrates, and polysaccharides that were not fermented in the rumen. The small intestine is not only a site of nutrient absorption but also helps balance immune function, stabilizes fermentation, and improves health and productivity [69].

A high prevalence of *Prevotella* in the rumen indicates a healthy microbiome [70]. *Prevotella* is a dominant genus in ruminants and is highly adaptable to the ruminal environment [71]. They play a key role in the degradation of cellulose, hemicellulose, and pectin [72]. Its main fermentation end product is propionic acid, which helps reduce methane production in the rumen [73], and is an important substrate for gluconeogenesis in the liver of ruminants [74]. *Rikenellaceae RC9 gut group* was the second most predominant genus increased in the YC, OA, and YO groups. Similar to our study, Zhang et al. [72] reported that *Rikenellaceae RC9 gut group* is one of the dominant genera in the rumen of grazing sheep. Cheng et al. [75] reported that *Rikenellaceae RC9 gut group* played a crucial role in the growth performance of Tan sheep. It can regulate the deposition of mutton fat by affecting the VFA concentration. Similarly, the growth performance results confirmed that fat thickness was significantly higher in the supplemented groups.

Supplementation with YC, OA, or YO increased *[Ruminococcus] gauvreauii group* in the small intestine. Su et al. [38] found that *[Ruminococcus] gauvreauii group* plays a significant role in hemicellulose and cellulose degradation. Cheng et al. [75] have reported a direct correlation between *Ruminococcus* spp. and body weight gain in lamb. It also confirmed that sheep supplemented with YC, OA, and YO had higher body weights.

Nucleotide metabolism, amino acid biosynthesis, aminoacyl-tRNA biosynthesis, and pyrimidine and purine metabolism are essential pathways for microbial protein synthesis and nutrient availability in the rumen [76]. A study on dairy cows showed that pyrimidine and purine metabolism are important for nitrogen metabolism in the rumen, accounting for 20% of the nitrogen obtained from these processes [77]. Animals can reuse pyrimidine nitrogen more efficiently than the nitrogen from purine metabolism [78]. Additionally, a study by Pedley and Benkovic [79] found that purine metabolism supplies the necessary components for RNA and DNA as well as energy for cell survival, growth, and proliferation in the body. Nucleotide metabolism is a key pathway in the rumen, where microbes convert dietary nitrogen, including nucleic acids, into microbial proteins, thereby contributing significantly to growth. This process accounts for a substantial portion of the total nitrogen utilized by ruminants [80]. Gao and Geng [81] reported that active yeast supplementation enhanced amino acid, lipid, carbohydrate, and energy metabolism in the rumen bacteria of finishing bulls. This study indicates that supplementing the diet with YC, OA, or YO can enhance growth performance by improving nitrogen utilization, microbial protein synthesis, and reducing harmful metabolites.

## 5. Conclusions

Supplementing a lamb diet with YC and OA improved the bacterial composition in the rumen and small intestine, nutrient digestibility, weight gain, carcass weight, serum biochemical indices, and antioxidant and calcium levels. It also enhances meat quality by improving color, fat content, and mineral availability and by regulating fatty acid levels in the muscle. Additionally, it increases the pH, volatile fatty acid concentration, and acetic to propionic acid ratio in the rumen. This supplementation enhances rumen function, growth, and fattening performance, thereby serving as a foundation for the development of novel supplements for ruminants. Future research should explore the long-term effects on sheep health and productivity, as well as the potential of YC to mitigate the antinutritional effects of oxalate-rich forage

## Figures and Tables

**Figure 1 microorganisms-13-02834-f001:**
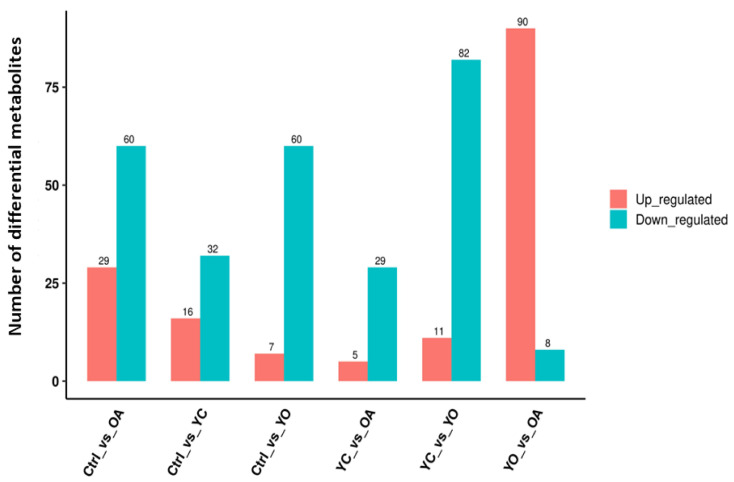
Differential metabolites in sheep serum across different comparison groups. ctrl: only a diet; YC: a diet with 0.625 g/kg yeast culture; OA: a diet with 0.4 g/kg oxalic acid; YO: a diet with 0.625 g/kg yeast culture and 0.4 g/kg oxalic acid (DM basis). A total of 4 treatment groups with 5 replications (*n* = 20) were analyzed. The x-axis represents the experimental group information, the y-axis represents the number of differential metabolites, and colors indicate up-regulated and down-regulated metabolites.

**Figure 2 microorganisms-13-02834-f002:**
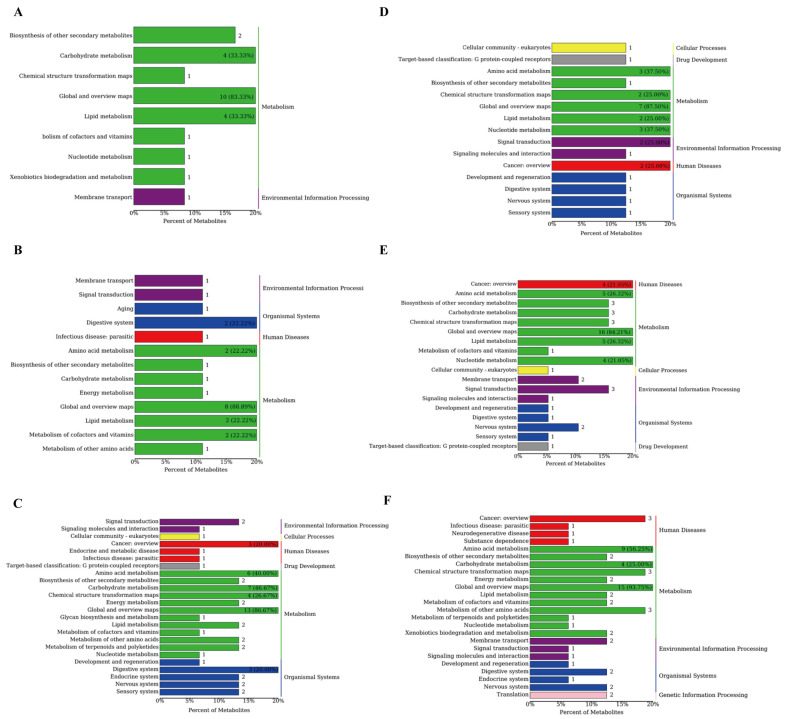
First (**right**) and second (**left**) level pathways with differential metabolites in serum. (**A**) ctrl and YC; (**B**) ctrl and OA; (**C**) ctrl and YO; (**D**) YC and OA; (**E**) YC and YO; (**F**) OA and YO groups. A total of 4 treatment groups with 5 replications (*n* = 20) were analyzed. The figure shows the secondary classification of KEGG pathways on the left and the primary classification on the right, color-coded for different primary classifications. The x-axis indicates the percentage of metabolites annotated to the secondary pathway out of the total annotated metabolites, with percentages over 20% labeled.

**Figure 3 microorganisms-13-02834-f003:**
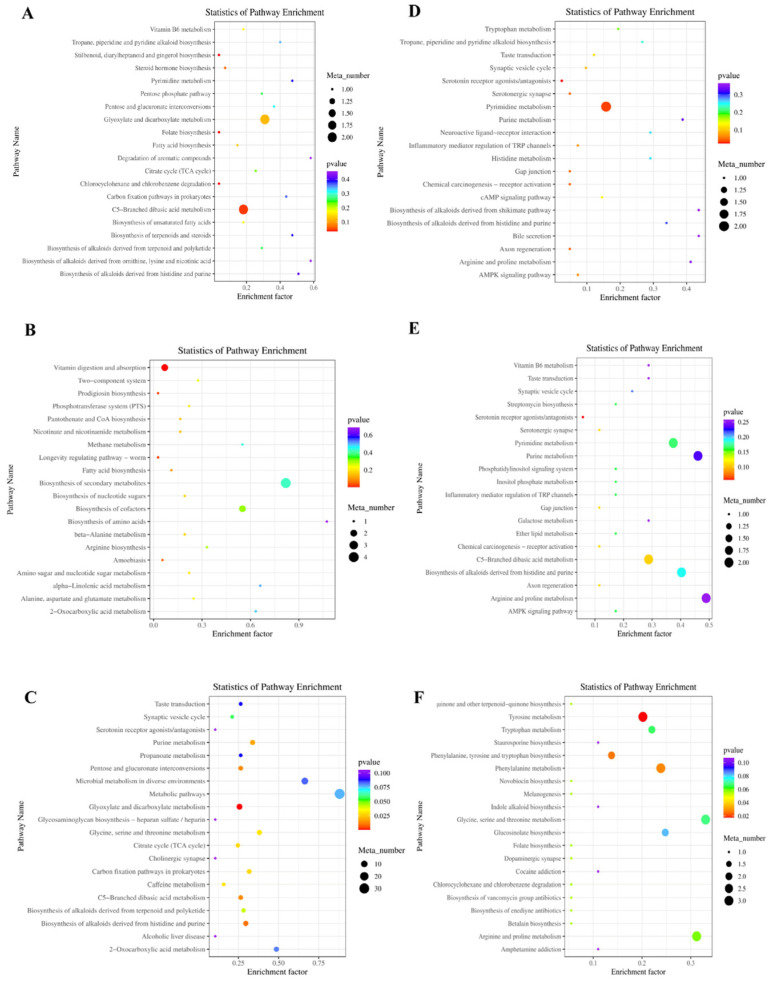
Third-level KEGG enriched pathways with differential metabolites in sheep serum. (**A**) ctrl and YC; (**B**) ctrl and OA; (**C**) ctrl and YO; (**D**) YC and OA; (**E**) YC and YO; (**F**) OA and YO groups. Included 4 experimental groups with 5 replications each (*n* = 20). It shows the top 20 pathways ranked by enrichment significance. The x-axis represents the enrichment factor, with a higher value indicating a more significant enrichment level of differential metabolites in the pathway. Dot color represents *p*-value, and bubble size indicates the number of differential metabolites in the pathway.

**Figure 4 microorganisms-13-02834-f004:**
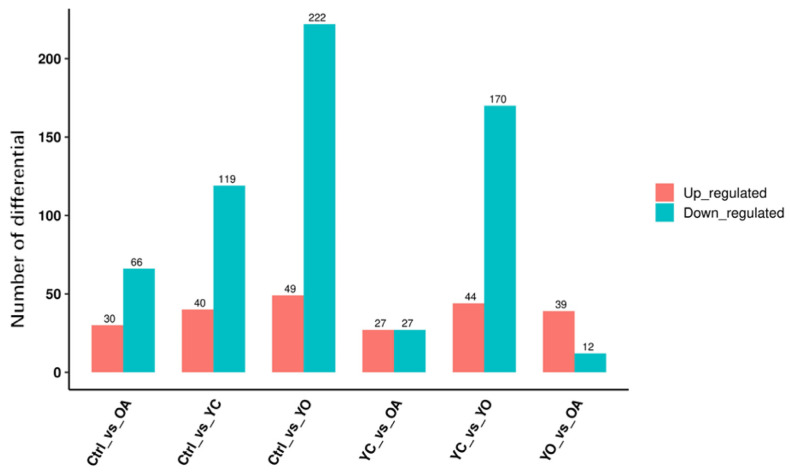
Differential metabolites in sheep longissimus dorsi muscle across different groups. ctrl: only a diet; YC: a diet with 0.625 g/kg yeast culture; OA: a diet with 0.4 g/kg oxalic acid; YO: a diet with 0.625 g/kg yeast culture and 0.4 g/kg oxalic acid (DM basis). A total of 4 treatment groups with 5 replications (*n* = 20) were analyzed. The x-axis represents the experimental group information, the y-axis represents the number of differential metabolites, and colors indicate up regulated and down regulated metabolites.

**Figure 5 microorganisms-13-02834-f005:**
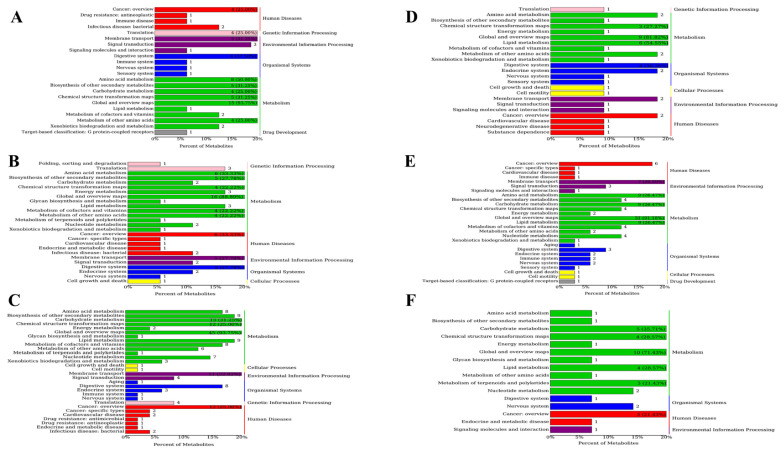
First (**right**) and second (**left**) level pathways with differential metabolites in longissimus dorsi. (**A**) ctrl and YC; (**B**) ctrl and OA; (**C**) ctrl and YO; (**D**) YC and OA; (**E**) YC and YO; (**F**) OA and YO groups. A total of 4 treatment groups with 5 replications (*n* = 20) were analyzed. The figure shows the secondary classification of KEGG pathways on the left and the primary classification on the right, color-coded for different primary classifications. The x-axis indicates the percentage of metabolites annotated to the secondary pathway out of the total annotated metabolites, with percentages over 20% labeled.

**Figure 6 microorganisms-13-02834-f006:**
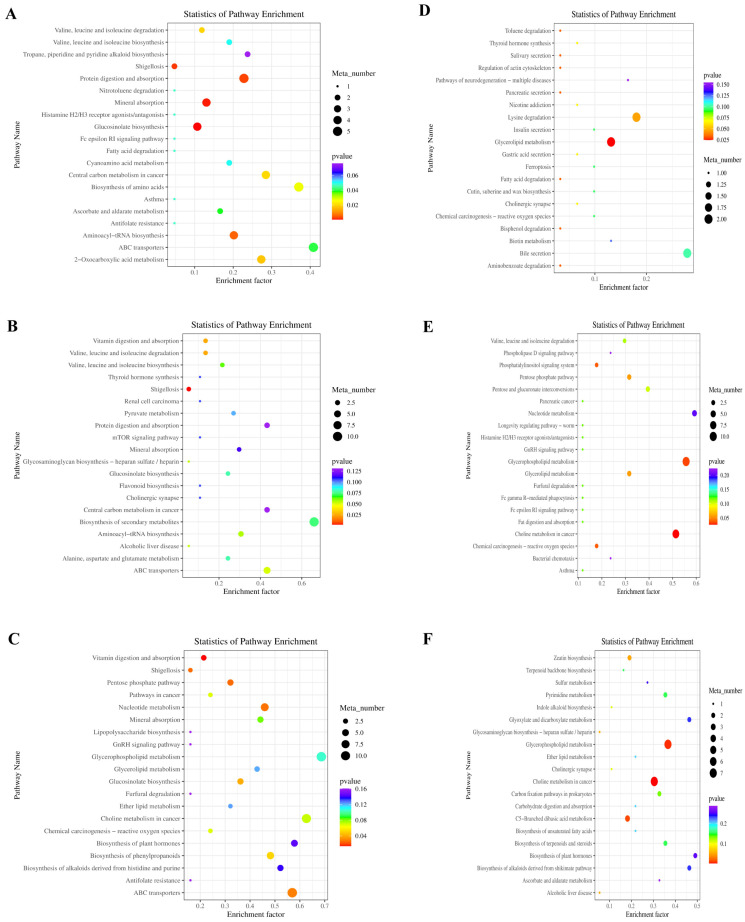
Third-level KEGG enriched pathways with differential metabolites in longissimus dorsi muscle between different groups. (**A**) ctrl and YC; (**B**) ctrl and OA; (**C**) ctrl and YO; (**D**) YC and OA; (**E**) YC and YO; (**F**) OA and YO groups. Included 4 experimental groups with 5 replications each (*n* = 20). It shows top 20 pathways ranked by enrichment significance. The x-axis represents the enrichment factor, with a higher value indicating a more significant enrichment level of differential metabolites in the pathway. Dot color represents *p*-value, and bubble size indicates the number of differential metabolites in the pathway.

**Figure 7 microorganisms-13-02834-f007:**
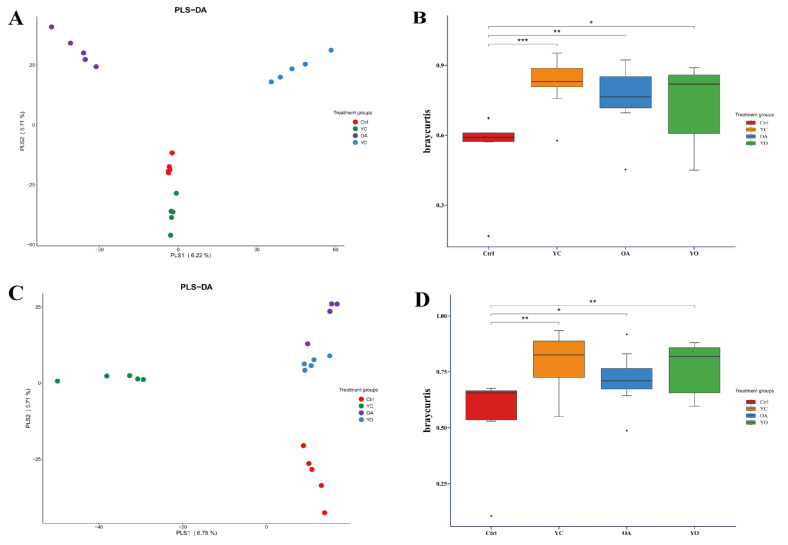
PLS-DA analysis of rumen (**A**) and the Bray–Curtis box showing distance differences between beta diversity groups for rumen (**B**); the PLS-DA analysis of small intestine (**C**); the Bray–Curtis box plot of distance differences between beta diversity groups for small intestine (**D**). ctrl: only a diet; YC: a diet with 0.625 g/kg yeast culture; OA: a diet with 0.4 g/kg oxalic acid; YO: a diet with 0.625 g/kg yeast culture and 0.4 g/kg oxalic acid (DM basis). A total of 4 treatment groups with 5 replications (*n* = 20) were analyzed. “*” indicating *p* ≤ 0.05, “**” for *p* ≤ 0.01, and “***” for *p* ≤ 0.001 on connecting lines.

**Figure 8 microorganisms-13-02834-f008:**
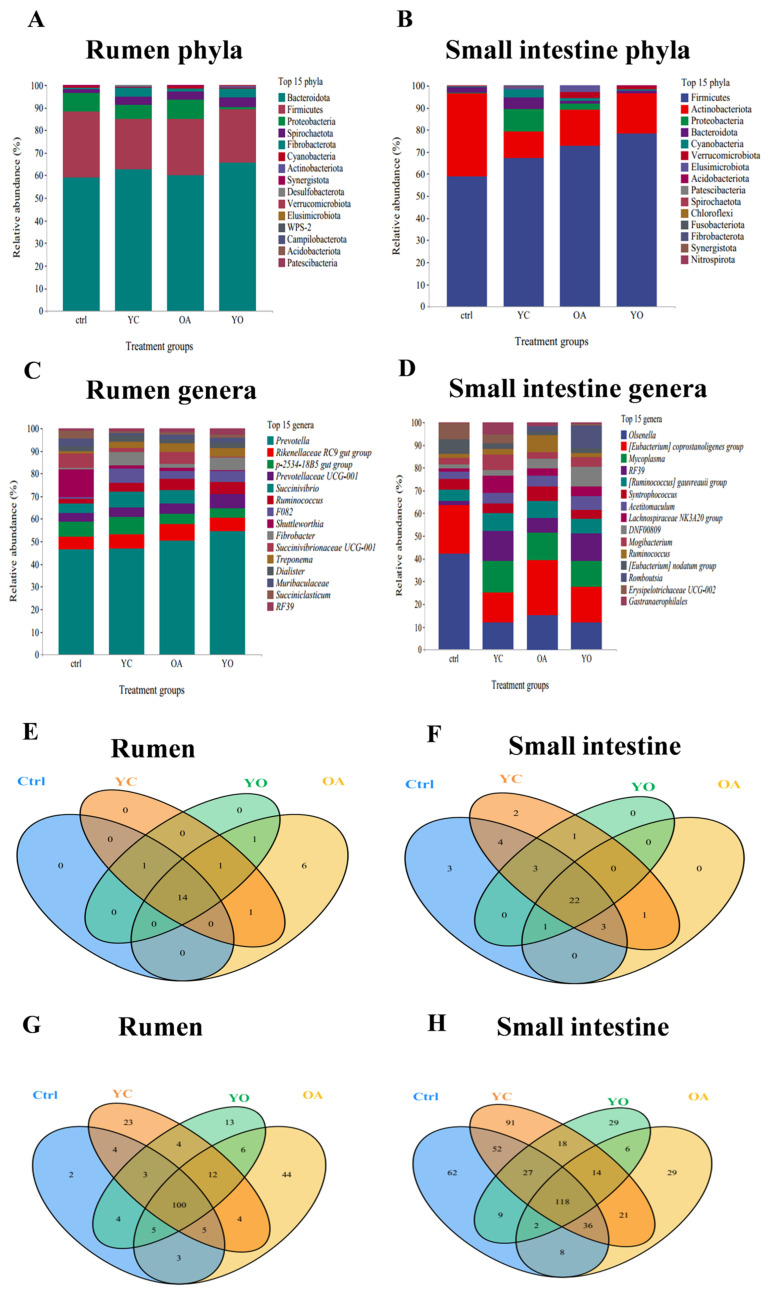
Bacterial composition in the rumen and small intestine. (**A**) rumen phylum composition, (**B**) small intestine phylum composition, (**C**) rumen genus composition, (**D**) small intestine genus composition, (**E**) phylum unique to each treatment group and those shared among groups in rumen, (**F**) phylum unique to each treatment group and those shared among groups in small intestine, (**G**) genus unique to each treatment group and those shared among groups in rumen, (**H**) genus unique to each treatment group and those shared among groups in small intestine. ctrl: only a diet; YC: a diet with 0.625 g/kg yeast culture; OA: a diet with 0.4 g/kg oxalic acid; YO: a diet with 0.625 g/kg yeast culture and 0.4 g/kg oxalic acid (DM basis). A total of 4 treatment groups with 5 replications (*n* = 20) were analyzed.

**Figure 9 microorganisms-13-02834-f009:**
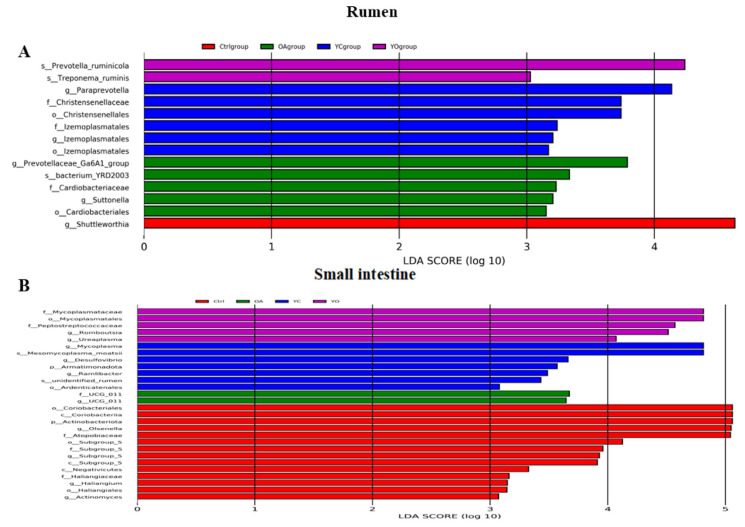
LEfSe analysis of taxa differences between groups for rumen (**A**) and small intestine (**B**). Data are shown as mean ± SEM. ctrl: only a diet; YC: a diet with 0.625 g/kg yeast culture; OA: a diet with 0.4 g/kg oxalic acid; YO: a diet with 0.625 g/kg yeast culture and 0.4 g/kg oxalic acid (DM basis). A total of 4 treatment groups with 5 replications (*n* = 20) were analyzed. The x-axis shows the logarithmic score of LDA for each taxon, whereas the y-axis represents the taxa. The LEfSe histogram highlights significantly enriched species and their effects within each group (*p* < 0.05, LDA score > 3).

**Figure 10 microorganisms-13-02834-f010:**
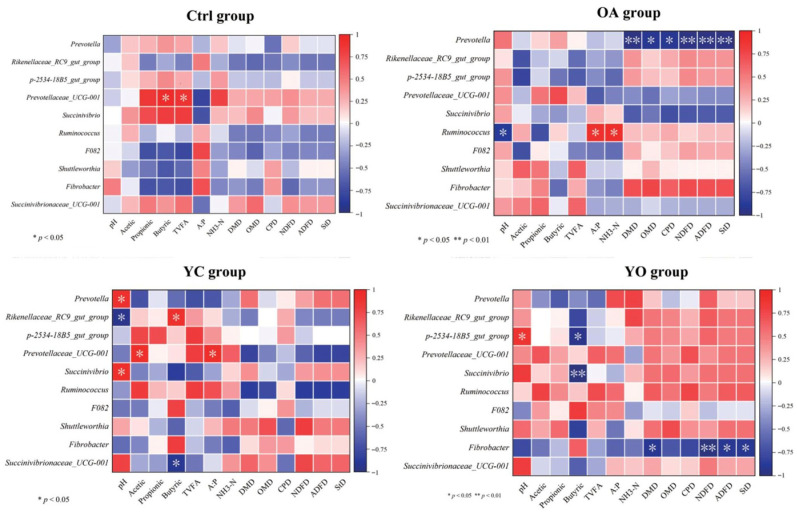
The correlation between specific bacterial genera in the sheep rumen and parameters related to rumen function and growth performance. * Represents *p* < 0.05 and ** *p* < 0.01. A total of 4 treatment groups with 5 replications (*n* = 20) were analyzed. TVFA: total volatile fatty acid; A/P: acetic to propionic acid ratio; NH_3_-N: ammonia nitrogen; DMD: dry matter digestibility; OMD: organic matter digestibility; CPD: crude protein digestibility; NDFD: neutral detergent fiber digestibility; ADFD: acid detergent fiber digestibility; and StD: starch digestibility. Red = positive correlation, blue = negative correlation.

**Figure 11 microorganisms-13-02834-f011:**
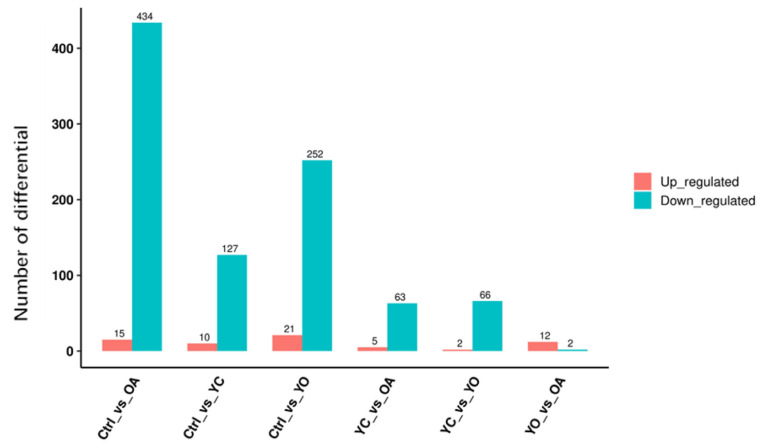
Differential metabolites in the rumen across experimental groups. ctrl: only a diet; YC: a diet with 0.625 g/kg yeast culture; OA: a diet with 0.4 g/kg oxalic acid; YO: a diet with 0.625 g/kg yeast culture and 0.4 g/kg oxalic acid (DM basis). A total of 4 treatment groups with 5 replications (*n* = 20) were analyzed.

**Figure 12 microorganisms-13-02834-f012:**
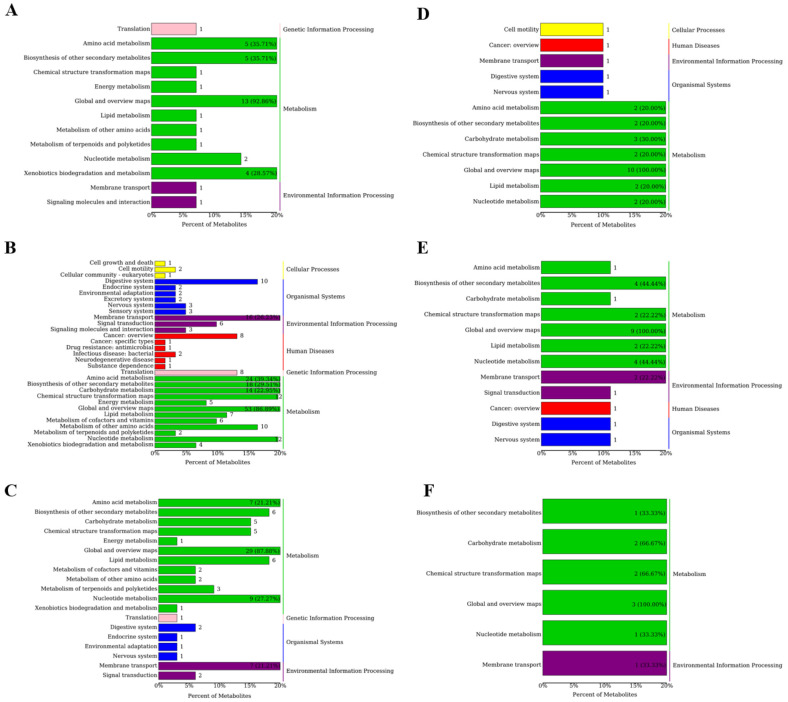
First (**right**) and second (**left**) level pathways with differential metabolites in rumen. (**A**) ctrl and YC; (**B**) ctrl and OA; (**C**) ctrl and YO; (**D**) YC and OA; (**E**) YC and YO; (**F**) OA and YO groups. A total of 4 treatment groups with 5 replications (*n* = 20) were analyzed. The figure shows the secondary classification of KEGG pathways on the left and the primary classification on the right, color-coded for different primary classifications. The x-axis indicates the percentage of metabolites annotated to the secondary pathway out of the total annotated metabolites, with percentages over 20% labeled.

**Figure 13 microorganisms-13-02834-f013:**
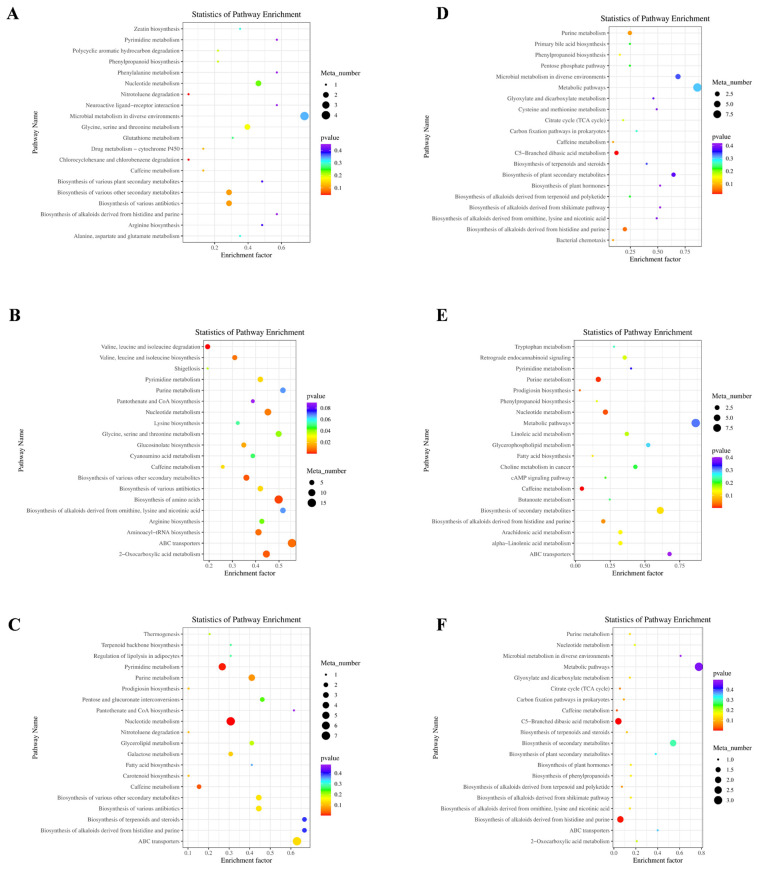
Differential metabolites identified in third-level KEGG pathways between treatment groups in rumen. (**A**) ctrl and YC; (**B**) ctrl and OA; (**C**) ctrl and YO; (**D**) YC and OA; (**E**) YC and YO; (**F**) OA and YO groups. Included 4 experimental groups with 5 replications each (*n* = 20). It shows top 20 pathways ranked by enrichment significance. The x-axis represents the enrichment factor, with a higher value indicating a more significant enrichment level of differential metabolites in the pathway. Dot color represents *p*-value, and bubble size indicates the number of differential metabolites in the pathway.

**Figure 14 microorganisms-13-02834-f014:**
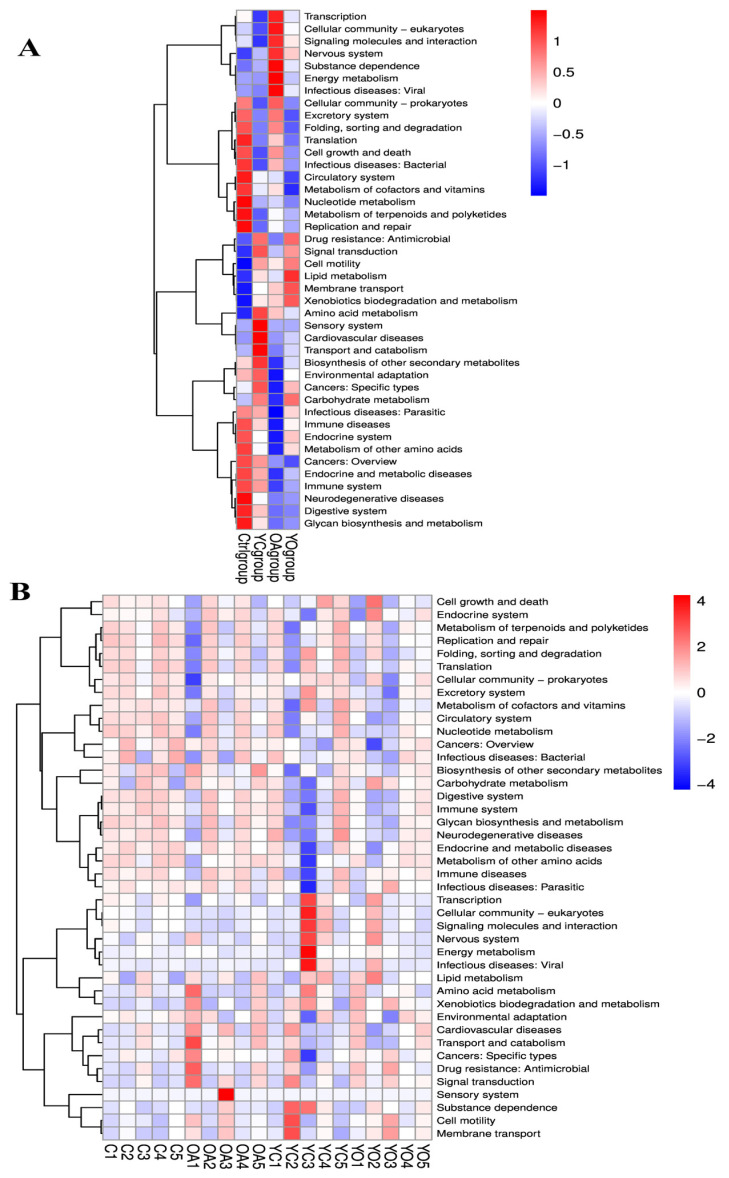
Functional predictions of KEGG 2nd level pathways in the rumen at group (**A**), and individual sample (**B**). Data are shown as mean ± SEM. ctrl: only a diet; YC: a diet with 0.625 g/kg yeast culture; OA: a diet with 0.4 g/kg oxalic acid; YO: a diet with 0.625 g/kg yeast culture and 0.4 g/kg oxalic acid (DM basis). A total of 4 treatment groups with 5 replications (*n* = 20) were analyzed.

**Table 1 microorganisms-13-02834-t001:** Feed ingredients and nutritional composition of the diet (% DM).

Ingredients		Chemical Composition	
Corn	35.71	Dry matter	90.30
Soybean meal	13.27	Metabolizable energy MJ/kg ^a^	10.00
Corn germ meal (spray)	6.12	Crude protein	13.90
Corn bran	31.63	Crude fat	2.80
Bentonite	4.08	Crude fiber	6.70
Sugar	4.08	Organic matter	87.60
Premix ^1^	5.10	Starch	30.30
Total	100.00	Neutral detergent fiber	25.80
		Acid detergent fiber	8.80
		Hemicellulose	17.00
		Calcium	0.60
		Phosphorus	0.40

^1^ Premix composition (per kilogram): 8.453 g FeSO_4_, 1.48 g CuSO_4_•5H_2_O, 13.241 g MnSO_4_, 8.294 g ZnSO_4_•5H_2_O, 0.016 g CoCl_2_, 0.03 g KI, 0.377 g Na_2_SeO_3_ (1% Se), 755 IU Vitamin A, 113 IU Vitamin D, and 887 IU Vitamin E. The hemicellulose content is calculated by subtracting the ADF from the NDF. The sheep were fed a diet consisting of pelleted feed as the main component and oat hay (*Avena sativa* L.) as a roughage. ^a^ Calculated according to NRC (2003).

**Table 2 microorganisms-13-02834-t002:** Effects of YC and OA supplementation on nutrient digestibility in sheep.

	Treatment Groups				SEM	*p*-Value		
Digestibility (%)	Ctrl	YC	OA	YO		YC	OA	YC × OA
Dry matter	75.63 ^b^	77.63 ^c^	77.99 ^c^	73.60 ^a^	0.468	0.040	0.130	<0.001
Organic matter	63.84 ^b^	64.17 ^bc^	64.54 ^c^	61.75 ^a^	0.263	<0.001	<0.001	<0.001
Crude protein	68.92 ^b^	72.28 ^c^	74.55 ^c^	64.55 ^a^	0.947	0.001	0.230	<0.001
Neutral detergent fiber	47.70 ^ab^	48.32 ^ab^	51.22 ^b^	45.08 ^a^	0.753	0.050	0.910	0.010
Acid detergent fiber	36.89 ^b^	42.05 ^c^	43.01 ^c^	31.63 ^a^	1.212	0.040	0.130	<0.001
Hemicellulose ^1^	63.56 ^b^	36.85 ^a^	48.29 ^a^	79.08 ^c^	4.103	0.625	0.005	<0.001
Starch	98.17 ^b^	98.32 ^c^	98.35 ^c^	98.02 ^a^	0.035	0.040	0.130	<0.001
Ash	59.34 ^b^	62.67 ^c^	63.28 ^c^	55.95 ^a^	0.781	0.040	0.130	<0.001
Ether extract	85.91 ^a^	90.15 ^c^	89.57 ^c^	88.06 ^b^	0.438	0.010	0.130	<0.001
Feces calcium (g/kg)	2.39 ^b^	2.10 ^a^	2.11 ^a^	2.17 ^a^	0.029	<0.001	<0.001	<0.001

^a, b, c^ Different superscripts within the same row indicate significant differences in mean values (*p* ≤ 0.05). ctrl: only a diet; YC: a diet with 0.625 g/kg yeast culture; OA: a diet with 0.4 g/kg oxalic acid; YO: a diet with 0.625 g/kg yeast culture and 0.4 g/kg oxalic acid (DM basis). A total of 4 treatment groups with 5 replications (*n* = 20) were analyzed. Hemicellulose ^1^ = NDF-ADF.

**Table 3 microorganisms-13-02834-t003:** Effects YC and OA supplementation on sheep growth performance.

Item	Treatment Groups				SEM	*p*-Value		
	Ctrl	YC	OA	YO		YC	OA	YC × OA
DMI (kg/d)	1.594 ^a^	1.72 ^c^	1.71 ^c^	1.69 ^ab^	0.019	0.111	0.210	0.050
ADG (kg/d)	0.248	0.266	0.28	0.26	0.012	0.530	0.990	0.671
BWG (kg)	15.20	16.40	15.60	15.90	0.509	0.871	0.981	0.361
FCR	6.432	6.400	6.762	6.452	0.195	0.580	0.770	0.870

^a, b, c^ Different superscripts within the same row indicate significant differences in mean values (*p* ≤ 0.05). ctrl: only a diet; YC: a diet with 0.625 g/kg yeast culture; OA: a diet with 0.4 g/kg oxalic acid; YO: a diet with 0.625 g/kg yeast culture and 0.4 g/kg oxalic acid (DM basis). A total of 4 treatment groups with 5 replications (*n* = 20) were analyzed. DMI: dry matter intake; ADG: average daily gain; BWG: body weight gain; FCR: feed conversion ratio.

**Table 4 microorganisms-13-02834-t004:** Effects YC and OA supplementation on sheep carcass characteristics.

Item	Treatment Groups				SEM	*p*-Value		
	Ctrl	YC	OA	YO		YC	OA	YC × OA
Hot carcass (kg)	20.90	22.70	22.20	21.70	0.355	0.340	0.820	0.100
Dressing percentage	48.71 ^a^	51.96 ^b^	50.30 ^ab^	49.99 ^ab^	0.430	0.070	0.800	0.030
Girth rib (GR) (mm)	5.53 ^a^	5.92 ^ab^	6.79 ^ab^	7.26 ^c^	0.265	0.370	0.010	0.930
Abdominal fat (kg)	0.71	0.81	0.79	0.81	0.060	0.630	0.760	0.760

^a, b, c^ Different superscripts within the same row indicate significant differences in mean values (*p* ≤ 0.05). ctrl: only a diet; YC: a diet with 0.625 g/kg yeast culture; OA: a diet with 0.4 g/kg oxalic acid; YO: a diet with 0.625 g/kg yeast culture and 0.4 g/kg oxalic acid (DM basis). A total of 4 treatment groups with 5 replications (*n* = 20) were analyzed.

**Table 5 microorganisms-13-02834-t005:** Effects of YC and OA on sheep serum indices and antioxidant levels.

Item	Day	Treatment Groups				SEM	*p*-Value		
		Ctrl	YC	OA	YO		YC	OA	YC × OA
TC (mmol/L)	0	2.80	2.500	2.72	2.83	0.06	0.405	0.287	0.097
	30	2.83	2.94	2.96	2.96	0.04	0.505	0.324	0.510
	60	2.72 ^b^	2.55 ^ab^	2.86 ^c^	2.23 ^a^	0.08	0.005	0.487	0.083
GLU (µmol/g)	0	4.87	4.66	4.90	4.74	0.10	0.374	0.788	0.904
	30	4.92	5.37	4.93	5.11	0.09	0.068	0.443	0.410
	60	5.07 ^ab^	5.49 ^b^	5.16 ^ab^	4.76 ^a^	0.11	0.971	0.113	0.047
BUN (mmol/L)	0	7.13 ^b^	6.60 ^b^	3.67 ^a^	5.99 ^b^	0.43	0.195	0.007	0.046
	30	8.33	7.54	7.92	7.49	0.24	0.235	0.648	0.719
	60	5.28 ^b^	7.36 ^c^	8.00 ^c^	3.22 ^a^	0.51	0.042	0.262	<0.001
IgA (g/L)	0	1.54	1.53	1.58	1.95	0.08	0.299	0.179	0.253
	30	1.65 ^a^	1.71 ^ab^	2.12 ^b^	2.02 ^ab^	0.08	0.917	0.008	0.542
	60	1.86	1.84	2.08	2.24	0.08	0.663	0.068	0.578
IgM (g/L)	0	4.06	4.01	3.48	3.96	0.13	0.434	0.252	0.336
	30	3.42	4.02	3.53	4.07	0.13	0.033	0.748	0.918
	60	3.47 ^a^	3.93 ^ab^	4.18 ^ab^	4.44 ^b^	0.15	0.207	0.041	0.722
sCa (g/L)	60	1.57 ^a^	1.44 ^a^	3.93 ^c^	2.97 ^b^	0.25	0.015	<0.001	0.053
sCr (umol/L)	60	67.99 ^c^	63.77 ^b^	70.15 ^c^	57.76 ^a^	1.14	<0.001	0.024	<0.001
SOD (ng/mL)	0	19.57	20.18	18.55	19.37	0.47	0.480	0.371	0.921
	30	21.08	22.53	21.63	23.14	0.65	0.290	0.673	0.985
	60	14.76 ^a^	20.31 ^b^	21.04 ^b^	21.10 ^b^	0.93	0.084	0.034	0.090
T-AOC (U/mL)	0	0.81	0.58	0.68	0.66	0.06	0.363	0.823	0.423
	30	0.77	0.44	0.73	0.60	0.06	0.061	0.574	0.397
	60	0.57 ^ab^	0.43 ^a^	0.67 ^b^	0.55 ^ab^	0.04	0.092	0.013	0.890
GH (ng/mL)	0	3.47 ^b^	3.70 ^b^	2.53 ^a^	2.48 ^a^	0.17	0.713	<0.01	0.573
	30	2.79 ^ab^	2.44 ^a^	2.60 ^a^	3.36 ^b^	0.13	0.384	0.132	0.027
	60	2.52 ^a^	3.08 ^ab^	2.71 ^ab^	3.61 ^b^	0.17	0.028	0.249	0.574

^a, b, c^ Different superscripts within the same row indicate significant differences in mean values (*p* ≤ 0.05). ctrl: only a diet; YC: a diet with 0.625 g/kg yeast culture; OA: a diet with 0.4 g/kg oxalic acid; YO: a diet with 0.625 g/kg yeast culture and 0.4 g/kg oxalic acid (DM basis). A total of 4 treatment groups with 5 replications (*n* = 20) were analyzed. TC: total cholesterol; GLU: glucose; IgA: immunoglobulin A; IgM: immunoglobulin M; BUN: blood urea nitrogen; sCa: serum calcium level; sCr: serum creatinine level; T-AOC: total antioxidant capacity; SOD: superoxide dismutase; GH: growth hormone.

**Table 6 microorganisms-13-02834-t006:** Effects on physicochemical properties of longissimus dorsi muscle in sheep.

Item	Treatment Groups				SEM	*p*-Value		
	Ctrl	YC	OA	YO		YC	OA	YC × OA
Brightness (L*)	26.92 ^a^	30.10 ^a^	34.20 ^b^	34.40 ^b^	0.885	0.161	<0.001	0.211
Redness (a*)	12.16 ^a^	15.41 ^c^	13.23 ^b^	12.80 ^b^	0.427	0.061	0.291	0.021
Yellowness (b*)	13.30 ^b^	12.27 ^b^	9.33 ^a^	8.52 ^a^	0.611	0.322	<0.001	0.912
Shear force (N)	70.30	72.74	73.01	75.83	1.333	0.352	0.311	0.950
Meat pH ^1^	6.82	7.25	7.17	7.14	0.075	0.751	0.421	0.900
Cooking loss (%)	33.87	30.45	32.25	30.95	0.739	0.128	0.706	0.479
Moisture (g/100 g)	72.26	71.20	71.98	71.58	0.191	0.058	0.890	0.369
Fat (g/100 g)	2.66 ^a^	4.02 ^b^	4.88 ^b^	4.06 ^b^	0.230	0.387	0.002	0.002
Protein (g/100 g)	22.42	22.40	21.64	21.92	0.191	0.737	0.118	0.699
Iron (mg/kg)	18.42 ^a^	17.12 ^a^	21.46 ^b^	20.78 ^b^	0.471	0.082	<0.001	0.569
Zinc (mg/kg)	30.64 ^a^	31.32 ^ab^	32.86 ^c^	32.72 ^bc^	0.306	0.576	0.002	0.399
Selenium (mg/kg)	0.088 ^ab^	0.094 ^b^	0.086 ^a^	0.086 ^a^	0.001	0.158	0.025	0.163
Phosphorus (g/kg)	1.93 ^c^	1.87 ^b^	1.82 ^b^	1.72 ^a^	0.019	<0.001	<0.001	0.317
Calcium (mg/kg)	70.92 ^c^	64.32 ^b^	65.92 ^b^	58.12 ^a^	1.125	<0.001	<0.001	0.508

^a, b, c^ Different superscripts within the same row indicate significant differences in mean values (*p* ≤ 0.05). pH ^1^: measured immediately after the slaughtering. ctrl: only a diet; YC: a diet with 0.625 g/kg yeast culture; OA: a diet with 0.4 g/kg oxalic acid; YO: a diet with 0.625 g/kg yeast culture and 0.4 g/kg oxalic acid (DM basis). A total of 4 treatment groups with 5 replications (*n* = 20) were analyzed.

**Table 7 microorganisms-13-02834-t007:** Effects on fatty acids composition of longissimus dorsi muscle in sheep (mg/g).

Item	Treatment Groups				SEM	*p*-Value		
	Ctrl	YC	OA	YO		YC	OA	YC × OA
C10:0 (Decanoic acid)	0.09	0.10	0.06	0.07	0.009	0.518	0.104	0.950
C11:0 (Undecanoic acid)	0.005 ^c^	0.005 ^bc^	0.003 ^a^	0.003 ^ab^	0.000	0.911	0.004	0.276
C12:0 (Lauric acid)	0.10 ^ab^	0.11 ^b^	0.06 ^a^	0.06 ^ab^	0.008	0.019	0.444	0.887
C13:0 (Tridecylic acid)	0.01 ^ab^	0.01 ^b^	0.004 ^a^	0.01 ^ab^	0.001	0.359	0.02	0.811
C14:0 (Myristic acid)	2.08 ^ab^	2.29 ^b^	1.05 ^a^	1.39 ^ab^	0.198	0.433	0.014	0.858
C14:1 (Myristoleic acid)	0.11 ^b^	0.10 ^b^	0.04 ^a^	0.06 ^ab^	0.010	0.607	0.011	0.542
C15:0 (Pentadecanoic acid)	0.26 ^ab^	0.31 ^b^	0.14 ^a^	0.18 ^ab^	0.025	0.324	0.010	0.99
C16:0 (Palmitic acid)	25.62 ^b^	26.69 ^b^	16.34 ^a^	17.24 ^a^	1.649	0.721	0.003	0.976
C16:1 (Palmitoleic acid)	2.05 ^b^	2.07 ^b^	1.02 ^a^	1.21 ^a^	1.588	0.706	0.002	0.752
C17:0 (Margaric acid)	0.90 ^ab^	1.03 ^b^	0.53 ^a^	0.59 ^a^	0.076	0.475	0.007	0.782
C17:1 (Heptadecenoic acid)	0.68 ^bc^	0.77 ^c^	0.36 ^a^	0.44 ^ab^	0.060	0.418	0.005	0.980
C18:0 (Stearic acid)	14.01 ^ab^	15.01 ^b^	10.30 ^a^	10.26 ^a^	0.834	0.747	0.011	0.727
C18:1n9c (Oleic acid)	31.93 ^ab^	38.63 ^b^	22.72 ^a^	23.45 ^ab^	2.684	0.455	0.023	0.546
C18:2n6c (Linoleic acid)	9.55 ^ab^	9.85 ^b^	7.43 ^ab^	6.93 ^a^	0.482	0.904	0.008	0.636
C20:0 (Arachidic acid)	0.07 ^b^	0.06 ^b^	0.05 ^a^	0.05 ^a^	0.003	0.917	<0.001	0.539
C18:3n6 (Omega 6 PUFA)	0.05	0.07	0.05	0.05	0.004	0.317	0.077	0.052
C20:1 (Eicosenoic acid)	0.08 ^ab^	0.10 ^b^	0.06 ^a^	0.07 ^ab^	0.005	0.267	0.015	0.685
C18:3n3 (α-linolenic acid)	0.18 ^ab^	0.22 ^b^	0.13 ^a^	0.13 ^a^	0.015	0.340	0.012	0.434
C20:3n6 (Homo- γ -Linolenic)	0.14	0.14	0.12	0.11	0.005	0.540	0.019	0.735
C23:0 (Tricosanoic acid)	0.028 ^b^	0.027 ^ab^	0.024 ^a^	0.023 ^a^	0.001	0.452	0.006	0.658
C24:1 (Neryonic acid)	0.07	0.06	0.06	0.12	0.020	0.492	0.520	0.314
∑SFA	43.16 ^ab^	45.65 ^b^	28.55 ^a^	29.90 ^a^	2.700	0.687	0.005	0.904
∑MUFA	34.93 ^ab^	41.73 ^b^	24.28 ^a^	25.35 ^a^	2.900	0.460	0.019	0.589
∑PUFA	9.92 ^ab^	10.28 ^b^	7.73 ^ab^	7.22 ^a^	0.500	0.930	0.008	0.619
PUFA/SFA	0.230 ^a^	0.23 ^a^	0.285 ^b^	0.24 ^ab^	0.010	0.238	0.062	0.230

^a, b, c^ Different superscripts within the same row indicate significant differences in mean values (*p* ≤ 0.05). ctrl: only a diet; YC: a diet with 0.625 g/kg yeast culture; OA: a diet with 0.4 g/kg oxalic acid; YO: a diet with 0.625 g/kg yeast culture and 0.4 g/kg oxalic acid (DM basis). A total of 4 treatment groups with 5 replications (*n* = 20) were analyzed. SFA: saturated fatty acids; MUFA: monounsaturated fatty acids; PUFA: polyunsaturated fatty acids; PUFA/SFA: polyunsaturated to saturated fatty acids ratio. ∑SFA = C10:0 + C11:0 + C12:0 + C13:0 + C14:0 + C15:0 + C16:0 + C17:0 + C18:0 + C20:0 + C23:0. ∑MUFA = C14:1 + C16:1 + C17:1 + C18:1n9c + C 20:1 + C24:1. ∑PUFA = C18:3n6t + C18:2n6c + C18:3n3 + C20:3n6.

**Table 8 microorganisms-13-02834-t008:** Effects on rumen fermentation parameters in sheep.

Item	Treatment Groups				SEM	*p*-Value		
	Ctrl	YC	OA	YO		YC	OA	YC × OA
Rumen pH	6.14 ^a^	6.48 ^ab^	6.58 ^b^	6.48 ^ab^	0.061	0.260	0.050	0.050
NH_3_-N (mg/dL)	14.55 ^a^	14.02 ^a^	13.12 ^a^	17.63 ^b^	0.463	0.002	0.070	<0.001
VFA (mmol/L)								
Acetic acid	22.58 ^a^	26.52 ^ab^	24.09 ^ab^	26.83 ^b^	0.716	0.021	0.490	0.650
Propionic acid	10.11 ^a^	12.02 ^b^	10.59 ^ab^	9.99 ^a^	0.331	0.291	0.210	0.050
Isobutyric acid	0.49	0.49	0.27	0.26	0.068	0.961	0.130	0.990
Butyric acid	2.29	1.73	1.82	2.41	0.330	0.980	0.890	0.420
Isovaleric acid	0.47	0.45	0.21	0.28	0.090	0.900	0.260	0.831
Valeric acid	0.71	0.72	0.45	0.40	0.080	0.910	0.060	0.841
TVFA ^3^	34.38 ^a^	42.28 ^b^	35.59 ^a^	39.43 ^ab^	1.081	0.004	0.646	0.271
TVFA ^6^	35.65 ^a^	43.65 ^b^	36.39 ^a^	40.36 ^ab^	1.144	<0.01	0.510	0.310
A/P	2.26 ^b^	1.92 ^a^	2.29 ^b^	2.70 ^c^	0.080	0.730	0.001	<0.01

^a, b, c^ Different superscripts within the same row indicate significant differences in mean values (*p* ≤ 0.05). ctrl: only a diet; YC: a diet with 0.625 g/kg yeast culture; OA: a diet with 0.4 g/kg oxalic acid; YO: a diet with 0.625 g/kg yeast culture and 0.4 g/kg oxalic acid (DM basis). A total of 4 treatment groups with 5 replications (*n* = 20) were analyzed. NH_3_-N: ammonia nitrogen; TVFA ^3^: total volatile fatty acids (acetic, propionic, and butyric acids), TVFA ^6^: total volatile fatty acids (acetic acid, propionic acid, isobutyric acid, butyric acid, isovaleric acid, and valeric acid); A/P: acetic to propionic acid ratio. We divided TVFA into TVFA^3^ and TVFA^6^ to specifically study the impact of supplementation on the key volatile fatty acids: acetic, propionic, and butyric.

**Table 9 microorganisms-13-02834-t009:** Effects on bacterial alpha diversity in the rumen and small intestine of sheep.

Fluid	A-Diversity Index	Treatment Groups				SEM	*p*-Value		
		Ctrl	YC	OA	YO		YC	OA	YC × OA
Rumen	Chao1	921.62 ^a^	1160.65 ^b^	1028.09 ^a^	1159.62 ^b^	45.854	0.048	0.551	0.543
	Good’s coverage	0.999	0.999	0.999	0.999	0.000	0.388	0.672	0.822
	Faith-pd	99.78 ^a^	120.26 ^b^	108.35 ^a^	119.74 ^b^	3.963	0.050	0.599	0.554
	Shannon	6.21 ^a^	6.58 ^ab^	6.44 ^ab^	6.79 ^b^	0.084	0.028	0.159	0.980
	Simpson	0.96 ^a^	0.96 ^a^	0.97 ^a^	0.98 ^b^	0.002	0.041	0.012	0.350
Small intestine	Chao1	720.83 ^ab^	807.76 ^b^	595.80 ^a^	566.75 ^a^	36.812	0.657	0.011	0.377
	Good’s coverage	0.997 ^a^	0.996 ^a^	0.998 ^b^	0.998 ^b^	0.000	0.832	0.003	0.917
	Faith-pd	60.27	65.80	55.06	52.59	2.345	0.735	0.054	0.38
	Shannon	5.11	5.45	5.65	5.43	0.133	0.827	0.357	0.31
	Simpson	0.91	0.91	0.95	0.94	0.011	0.810	0.142	0.923

^a, b^ Different superscripts within the same row indicate significant differences in mean values (*p* ≤ 0.05). ctrl: only a diet; YC: a diet with 0.625 g/kg yeast culture; OA: a diet with 0.4 g/kg oxalic acid; YO: a diet with 0.625 g/kg yeast culture and 0.4 g/kg oxalic acid (DM basis). A total of 4 treatment groups with 5 replications (*n* = 20) were analyzed.

## Data Availability

The National Center for Biotechnology Information (NCBI) Sequence Read Archive (SRA) database stores raw sequence reads for samples in this study (PRJNA1094447).

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
