# Peer review of "Enhancing Sheep Rumen Function, and Growth Performance Through Yeast Culture and Oxalic Acid Supplementation in a Hemicellulose-Based Diet"

_microorganisms, 2025, doi:10.3390/microorganisms13122834_

Round 1

Reviewer 1 Report

Comments and Suggestions for Authors

Overall, the manuscript is well-written and provides comprehensive information regarding the effects of oxalic acid and yeast culture. However, several issues must be addressed to improve the document.

First, I found that this manuscript is currently available as an uploaded preprint in another journal without peer review: (https://papers.ssrn.com/sol3/papers.cfm?abstract_id=4915037). The authors must clarify whether the manuscript is under consideration elsewhere, in accordance with the journal’s ethical policies.

The title is too long, and some terms are repetitive (e.g., growth and fattening performance). A review is recommended to shorten and refine the title.

In the abstract, all scientific names must be italicized (this observation applies to the entire manuscript). All abbreviations (DM, OM, CP, NDF) must be specified before. Additionally, the experimental period mentioned in the abstract does not match the one reported in the Materials and Methods section (75 d vs. 60 d).

Several sections of the Materials and Methods are difficult to follow. A thorough revision of the sequential order in the following sections is recommended:
“2.2 Experimental treatments and animal care,” “2.7 Slaughter samples,” and “2.8 Rumen and intestinal bacterial community.” It is also recommended to separate the description of sample collection and analysis for fermentation profile and meat quality in section 2.7 Slaughter samples.

In the Materials and Methods section, the author does not mention the composition, reagent characteristics, supplier information, variables evaluated, or describe how to calculate a value or the technique used. Please check: L134, L158, L144-152, L160-171, L194-196, L205, L209, L184-217 (meat quality variables).

The statistical analysis section is poorly written. The authors must describe clearly how each variable was analyzed. Was the same statistical model used for all analyses? Moreover, the manuscript lacks a description of the statistical methods used for the bioinformatics data.

The interpretation of the results must be substantially revised. This is a factorial design; only the main effects should be discussed when they are significant, and no interaction is detected. And when the interaction is significant, the discussion should focus solely on the interaction. In addition, the authors frequently describe “tendencies” or significant changes that are not supported by the reported p-values. Several sentences must be corrected: L188, L280-282, L284, L285, L295, L278-279, L310, L397, L509, and L525. Finally, the assignment of superscripts must be reviewed carefully, particularly in Tables 2, 3, 5, and 6.

The authors must specify which statistical analyses were used to detect differences in: “L558 Bacterial phylum composition in the rumen and small intestine” y “L575 Bacterial genus composition in the rumen and small intestine”, and “L628 Correlations between bacterial genera and rumen fermentation and growth in sheep”.

The discussion is weak. The authors do not provide sufficient evidence to support their findings, and the research question remains partially unaddressed. The potential mechanisms of YC and OA should be clearly linked to the results to improve the reader’s understanding.

Specific observations

L63-67: The sentence is confusing, as the described effects of oxalic acid seem contradictory. It is unclear how OA poses a risk to ruminants and how YC would mitigate these effects. I suggest including studies evaluating the individual effects of OA and YC.

L104: Writing “(mean ± SD)” is unnecessary. Replace “acclimated” with “adapted.”; and L892, replace “Lamb” with “sheep”.

L134-136, L160-171: Explanation is difficult to follow, and the purpose of the prepared solutions is unclear. The paragraphs must be rewritten.

L188: Replace “ruminal fluids” with “ruminal liquid.” Was a pooled sample (rumen and intestine) used? Is pooling valid for ruminal fermentation profile evaluation? How much ruminal and intestinal liquid was collected, and how much was analyzed for VFA and NH₃-N? Specify the concentration of metaphosphoric acid solution.

L184-217, section 2.7 Slaughter samples”: The anatomical location where fat thickness and girth rib were measured must be specified.

L288, Table 3: I recommend separating the Growth performance and carcass characteristics.

L300: Avoid interpretation in the results section: “indicating potential immune system enhancement in sheep”.

L337-347: This information appears to be duplicated in Figure 1.

L515, Table 7: Explain why TVFA is separated into TVFA3 and TVFA6.

L547: Correct the figure number.

Author Response

Reviewer 1

General observations

  1. First, I found that this manuscript is currently available as an uploaded pre-print in an other journal without peer review. The authors must clarify whether the manuscript is under consideration elsewhere, in accordance with the journal’s ethical policies.  

Response: Dear reviewer, Yes, the pre-print was uploaded by the Animal Nutrition journal, but it will not be published in the journal. We wanted to inform you that the manuscript is not currently being considered for publication elsewhere.

  1. The title is too long and some terms are repetitive (e.g. growth performance and fattening performance). A review is recommended to shorten and refine the title.

Response: The title is shorten as “Enhancing sheep rumen function, and growth performance through yeast culture and oxalic acid supplementation in a hemicellulose-based diet”

  1. In the abstract, all scientific names be italicized (this observation applies to the entire manuscript). All abbreviations (DM, OM, CP, and NDF) must be specified before. Additionally, the experimental period mentioned in the abstract does not match the one reported in the materials and methods section (75 d vs. 60 d).

Response: All the scientific names are italicized and the abbreviations in the abstract are defined (Lines: 33-35). The duration of the experiment mentioned in the abstract and materials and methods is similar. In the materials and methods section, it is stated that the study lasted for a total of 75 days, with 15 days allocated for adaptation and 60 days for the experimental period. Based on your direction and since it refers the exact numbers of experimental days, we corrected it 60 d in the abstract (line: 30).

  1. Several sections of the materials and methods are difficult to follow. A thorough revision of the sequential order in the following section is recommended: “2.2 Experimental treatments and animal care“, “ 2.7 Slaughter samples,” and “  2.8 Rumen and intestinal bacterial community.” It is also recommended to separate the description sample collection and analysis for fermentation profile and meat quality in section 2.7 slaughter samples.

Response: Dear reviewer, we have reorganized the text to align with the subtitle orders in sections 2.2 and 2.7. Furthermore, section 2.7 has been divided into subsections 2.7.1 (Meat quality traits) and 2.7.2 (Fermentation profiles). Lines: 101-105, 208-212.

  1. In the Materials and Methods section, the author does not mention the composition, reagent characteristics, supplier information, variables evaluated, describe how to calculate a value or the technique used. Please check L134, L158, L144-152, L160-171, L194-196, L205, L209, L184-217 (meat quality variables).

Response: Dear reviewer, we calculated the digestible energy values of the diet ingredients using data from a feed database in China, considering the variations in digestible energy values across different agro-ecologies. Serum creatinine and calcium levels were analyzed through Qingdao Stantec Standard Testing Co. Ltd. using an inductively coupled plasma mass spectrometer (ICPMS) biomedical instrument. To measure Serum creatinine concentration, the process involved diluting the serum with an acidic solution and adding an internal standard. The sample was then analyzed in the ICP-MS, where the plasma components were converted into ions. The instrument detected specific signals derived from creatinine and compared them with calibrated standards. Signal intensities were used to calculate creatinine concentration, and accuracy was verified using quality-control samples. Calcium analysis involved digesting the sample with ultrapure nitric acid, adding an internal standard (aluminum), and diluting the sample for analysis. Collision/reaction cells with gas such as helium was used to minimize interferences for specific calcium isotopes. Calcium analysis involved digesting the sample with ultrapure nitric acid, adding an internal standard (aluminum), and diluting the sample for analysis. Collision/reaction cells with gas such as helium was used to minimize interferences for specific calcium isotopes. Lines: 169-178, 180-194.

We also revised other sentences accordingly.

  1. The statistical analysis is poorly written. The authors must describe clearly how each variable was analyzed. Was the same statistical model used for all analysis? Moreover, the manuscript lacks a description of the statistical methods used for the bioinformatics data.

Response: The sequencing data was processed through various steps, such as sorting samples by barcodes, merging paired reads, quality control, and filtering to ensure high-quality sequences. Data demultiplexing was done using specific mismatch criteria for barcodes and primers. Pandaseq software was used to merge paired reads with defined overlap and mismatch parameters [25]. PRINSEQ software was employed to filter out low-quality bases and sequences with excessive N bases [26]. Metabolomics data analysis was conducted using LC-MS/MS (Orbitrap Exploris 120, Thermo Fisher Scientific, USA) [27] with three main components: basic analysis, advanced analysis, and personalized analysis. Basic analysis includes univariate and multivariate statistical analysis to identify significant metabolite differences. Personalized analysis involves bioinformatics analysis of these metabolites. Advanced analysis was customized to meet specific requirements, including multi-omics association analysis and the creation of custom charts for publications [28]. Lines: 297-309.

  1. The interpretation of the results must be substantially revised. This is a factorial design; only the main effects should be discussed when they are significant, and no interaction is detected. And when the interaction is significant, the discussion should focus solely on the interaction. In addition, the authors frequently describe “tendencies” or significant changes that are not supported by the reported p-values. Several sentences must be corrected: L188, L280-282, L284, L285, L295, L278-279, L310, L397, L509, and L525. Finally, the assignment of superscripts must be reviewed carefully, particularly in Tables 2, 3, 5, and 6.

Response: We have revised the result based on the recommendations.

  1. The authors must specify which statistical analyses were used to detect differences in “L558 bacterial phylum composition in the rumen and small intestine” L575 Bacterial genus composition in the rumen and small intestine” and L628 correlation between bacterial genera and rumen fermentation and growth in sheep”.

Response: Dear reviewer, the bacterial composition in the rumen and small intestine was analyzed using alpha and beta diversity indices with different software tools. Alpha diversity analysis was performed using QIIME2, including dilution curve analysis in R (version 3.6.2) [23]. Species richness and diversity were evaluated using the Chao1, Shannon, and Simpson indices. Beta diversity analysis was performed using QIIME2 to compare the species diversity among the samples. PLS-DA was used to investigate differences in the microbial community structure and the impact of taxonomies on the samples. To evaluate the correlation between bacterial genera and rumen fermentation and growth, we performed canonical correlation analysis using SPSS version 28 (Armonk, NY: IBM Corp).

  1. The discussion is weak. The authors do not provide sufficient evidence to support their findings, and the research question remains partially unaddressed. The potential mechanisms of YC and OA should be clearly linked to the results to improve the reader’s understanding.

Response: Dear reviewer, we have made some amendments to enhance the discussion section.

Specific observations

  1. L63-67: The sentence is confusing, as the described effect of oxalic acid seem contradictory. It is unclear how OA poses a risk to ruminants and how YC would mitigate these effects. I suggest including studies evaluating the individual effects of OA and YC.

Response: Oxalic acid, when combined with other ingredients, forms oxalate in tropical forages which can negatively impact the health of ruminants by binding to essential minerals such as Ca, Mg, and trace minerals such as Fe, rendering them unavailable to the body  [8]. High levels of oxalate in the blood can lead to the formation of crystals that can block urine flow and potentially cause kidney failure [7]. However, We used pure oxalic acid from a reagent-producing company. Lines: 75-80.

  1. L104: Writing (mean±SD) is unnecessary. Replace “acclimated” with “adapted”

Response: we removed the mean ± SD and replaced "acclimated" with "adapted." (L115-116)

  1. L134-136, L160-171: explanation is difficult to follow, and the purpose of the prepared solutions is unclear. The paragraphs must be rewritten.

Response: Dear reviewer, we calculated the digestible energy values of the diet ingredients using data from a feed database in China, considering the variations in digestible energy values across different agro-ecologies. we have revised the text to enhance clarity and readability. Lines: 144-148.

  1. L188, replace “ruminal fluid” with “ruminal liquid.” Was a pooled sample (rumen and intestine) used? Is pooling valid for rumen fermentation profile evaluation? How much ruminal and intestinal liquids was collected, and how much was analyzed for VFA and NH3-N? Specify the concentration of metaphospheric acid solution.

Response: Dear reviewer, we collected ruminal and intestinal liquids from various parts of the rumen and intestine, which is why we referred to them as pooled samples. We collected ruminal liquid from each sheep in one 5 mL centrifuge tube and four 2 mL centrifuge tubes, along with intestinal liquid in three 2 mL centrifuge tubes. The concentration of metaphosphoric acid solution was 2.5% (w/v).  Lines: 241-242.

  1. L184-217, section 2.7 slaughter samples the anatomical location where fat thickness and Girth rib were measured must be specified.

Response: We measured back fat thickness between the twelfth and thirteenth ribs on the left side of the carcass using a vernier caliper and calculated the girth rib (GR) values. The GR measurement is taken by measuring the circumference around the sheep's chest, just behind the front legs and withers. Lines: 209-213.

  1. L288, Table 3: I recommend separating the growth performance and carcass characteristics.

Response: we separated Table 3 into two as Table 3 (growth performance and Table 4 (carcass characteristics). Line: 338.

  1. L300: Avoid the interpretation in the result section “indicating potential immune system enhancement in sheep”

Response: we removed the phrase “indicating potential immune system enhancement in sheep”.

  1. L337-347: this information appears to be duplicated in Figure 1

Response: we have removed the duplication and only YO vs. OA has been mentioned.

  1. L 515, Table 7: explain why TVFA is separated into TVFA3and TVFA6

Response: We divided TVFA into TVFA3 and TVFA6 to specifically study the impact of supplementation on the key volatile fatty acids: acetic, propionic, and butyric. Lines: 568-569.

  1. Line 547 correct the figure number

Response: Dear reviewer, the figure number has been updated. Figure 8C has been replaced with Figure 7C.

Dear reviewers and editor, we appreciate your feedback and suggestions.

Reviewer 2 Report

Comments and Suggestions for Authors

The manuscript submitted for review addresses the still relevant issue of supplementing the diet of farm animals with compounds generally aimed at increasing productivity, health, etc.

My most important comment concerns the length of the manuscript /  the number of results presented. I suggest considering dividing the results into two parts and presenting them in separate papers, e.g., separating the results concerning meat quality and production parameters. This would make the text clearer and allow for a more thorough discussion, for example, of issues related to fatty acids, which are currently only mentioned in the discussion. 

My other comments are as follows:

line 47 - Is the cost of small ruminants, liek sheep, feeding such a big pronblem like in the case of bigger animals, often kept in different, intensive maintenance systems?

lines 63 - 71 - Is there any risk that oxalic acid as a feed supplement can form oxalate, which megativelu affects the health of the animals, as you mentioned in lines 64-65?

lines 76-77 - In which was this may mitigate the risk?

line 113 - How were the fecal samples collected?

line 135 p- Were feces collecyed individually from each animal? What is the direct collection method?

Author Response

Reviewer 2

My most important comment concerns the length of the manuscript / the number of results presented. I suggest considering dividing the results into two parts and presenting them in separate papers, e.g. separating the results concerning meat quality and production parameters. This would make the text clear and allow for a more thorough discussion, for example of issues related to fatty acids, which are currently only mentioned in the discussion.

Response: Dear reviewer, thank you for the suggestion to split the manuscript into two papers. However, we believe that combining the content into one strong manuscript would be more effective.

My other comments are as follows:

  1. Line 47- is the cost of small ruminants, like sheep, feeding such a big problem like in the case of bigger animals, often kept in different, intensive maintenance system?

Response: Dear reviewer, Small ruminant nutrition can be expensive due to the need for high-quality feed, supplements, and proper management to ensure health, growth, and productivity, particularly during times of scarcity or seasonal feed shortages.

  1. Lines 63-71- is there any risk that oxalic acid as a feed supplement can form oxalate, which negatively affects the health of the animals, as you mentioned in lines 64-65?

Response: Oxalic acid, when combined with other ingredients, forms oxalate in tropical forages which can negatively impact the health of ruminants by binding to essential minerals such as Ca, Mg, and trace minerals such as Fe, rendering them unavailable to the body  [8]. High levels of oxalate in the blood can lead to the formation of crystals that can block urine flow and potentially cause kidney failure [7]. Lines: 72-77.

  1. Lines 76-77- in which way this may mitigate the risk?

Response: Supplementing with minerals like calcium and magnesium can help reduce the risk of oxalic acid by binding oxalates in the gut, supporting oxalate-degrading microbes in the rumen, preventing mineral deficiencies, and reducing kidney crystal formation. Lines: 77-80.

  1. Line 113- how were the fecal samples collected?

Response: Sheep were housed individually, and their feces were manually collected to ensure precise measurement of total output for digestibility studies.

  1. Line 135 p- were feces collected individually from each animal? What is the direct collection method?

Response: Yes, it was collected from each animal individually for the last 5 experimental days. The direct method of fecal collection involves collecting all feces excreted by an animal as soon as it is voided, without using markers or estimation.

Dear reviewers and editor, we appreciate your feedback and suggestions.

Round 2

Reviewer 1 Report

Comments and Suggestions for Authors

The authors addressed the observations and recommendations in this revised version.

Reviewer 2 Report

Comments and Suggestions for Authors

Thank you for your answers and additional information, I am satisfied.